# Activation Method and Reuse of Waste Concrete Powder—A Review

Changming Bu [1,2], Baolin Tan [1,2], Qiutong Wu [1,2], Yingying Qiao [1,2], Yi Sun [1,2,*], Linwen Yu [3,*] and Qian Yang [1,2]

1   School of Civil Engineering and Architecture, Chongqing University of Science & Technology, Chongqing 401331, China; buchangming@cqust.edu.cn (C.B.); 2021206114@cqust.edu.cn (B.T.); 2021206120@cqust.edu.cn (Q.W.); 2022206055@cqust.edu.cn (Y.Q.); 2022206048@cqust.edu.cn (Q.Y.)
2   Chongqing Key Laboratory of Energy Engineering Mechanics & Disaster Prevention and Mitigation, Chongqing 401331, China
3   School of Materials Science and Engineering, Chongqing University, Chongqing 400044, China
*   Correspondence: sunyi@cqust.edu.cn (Y.S.); linwen.yu@cqu.edu.cn (L.Y.); Tel.: +86-135-9416-9610 (Y.S.)

**Abstract:** With an emphasis on environmental protection and the sustainable development of resources, the reuse of waste concrete has long been a research hotspot, and the study of WCP is the key to improving the efficiency of waste concrete utilization. In this study, in which we reviewed the relevant literature at home and abroad in recent years, we first used Citespace software to visualize and analyze the research on the reuse and activation methods of WCP in recent years. In this paper, we explain the characteristics of WCP and the influence of different activation methods on the activity index of WCP. We summarize the mechanical properties and working properties of WCP mortar products, and finally, the optimal activation method of WCP and the optimal amount of WCP in mortar preparation are analyzed. In addition, some problems in the current research are determined.

**Keywords:** waste concrete powder; activation; mortar; reuse; Citespace





## 1. Introduction

With accelerated urbanization, the annual production of construction waste exceeds 3.5 billion tons worldwide, which poses great challenges for the landfill disposal of construction waste and environmental protection [1,2]. Waste mortar and waste concrete constitute the bulk of construction waste, and waste concrete, as a recyclable resource, is an effective way to treat waste concrete by crushing it into a recycled aggregate (RCA) and powder (WCP) for replacing natural aggregates and cement, respectively [3,4]. Currently, most of the studies on waste concrete have focused on the preparation of recycled aggregates, and fewer studies have focused on waste concrete powder (WCP).

Cement-based materials, as commonly used cementitious materials, are widely used in the construction industry due to the easy availability and low cost of raw materials. Currently, the world produces 4.2 billion tons per year of silicate cement, which is the largest man-made material in the world, and 7% of the annual global $CO_2$ emissions come from cement production [5,6]. Excessive $CO_2$ emissions have caused many global problems, such as global warming, sea level rise, and glacier melting [7]. It is imperative to save energy and reduce $CO_2$ emissions [8]. Cui [7] used sulfoaluminate cement instead of normal silicate cement to prepare low-carbon eco-ultra-high-performance concrete. WCP, as a by-product of the waste concrete disposal process, has a similar chemical composition to cement, and the use of WCP to replace some cement in mortar and concrete is environmentally friendly and effective [9].

WCP cannot be widely used in construction due to its low activity and huge water requirement, which leads to its need for active stimulation [10]. Sui [11] found that WCP generates active substances such as dacite ($\beta$-$C_2S$) and C-S-H gel during heat treatment;

Li [12] found that WCP can be effectively activated under an alkaline environment to generate a large amount of C-S-H gels. In addition, mechanical grinding can also stimulate the activity of WCP, Yang [13] found that the activity of WCP after 1 h of ball milling was effectively stimulated, and Deng [14] proposed that submicron WCP produced by the wet grinding method can effectively improve the early strength and hydration of concrete. Due to the large amount of CaO within WCP, some scholars used $CO_2$ to treat it. Cheng [15] found that mortar mixed with carbonated WCP had higher flow and compressive strength than mortar without carbonated WCP. In addition, some scholars have improved the hydration of mortar and concrete by WCP compounded by external admixtures. Chen Xi [16] found that the mixture of WCP, fly ash (FA), and silica fume (SF) could replace 15% of the cement in mortar specimens and could improve the 28-day strength of mortar specimens. Ma [17] found that active powder (WP) mixed using WCP and waste brick powder could accelerate the hydration reaction of cementitious materials, and the addition of some WP could improve the pore structure of cementitious materials. Some scholars [18–20] have found that replacing cement with WCP had negative effects on the mechanical properties and early cracking of the products but could reduce the drying shrinkage of the products. Since the activation principles of mechanical activation, chemical activation, and thermal activation are different, their activation effects are not the same.

This paper describes the activation methods and applications of WCP and details the effects of different excitation methods on the WCP activity index. The mechanical and working properties of WCP products are also summarized. In addition, Citespace software was used to visualize and analyze the activation methods and reuse studies of WCP in recent years. Finally, some reflections and ideas are presented.

## 2. Properties of Waste Concrete Powder

The production process of recycled aggregates inevitably produces micronized particles (d < 150 µm), which account for about 19% of the mass of waste concrete, i.e., waste concrete powder (WCP) [21]. WCP is a fine, loose, and irregularly shaped powder [22] that is off-white, as shown in Figures 1 and 2, with a bulk density of 855–917 kg/m$^3$, an apparent density of 2355–2651 kg/m$^3$, and a stable specific surface area of 450–500 m$^2$/kg [19,20,23]. The main chemical composition of WCP in some of the representative literature is summarized in Table 1, in which it can be seen that the different sources of WCP lead to complex compositions and large variations in the contents, but its main components are the same as those of cement. The XRD patterns of WCP are shown in Figure 2. In the figure, it can be seen that the main mineral components of WCP are $SiO_2$, $CaCO_3$, $CaMg(CO_3)_2$, etc., all of which have potential activation properties. Therefore, certain methods can be used to stimulate the activity of WCP to make it a cementitious material with high hydration properties [24]. Compared to silicate cement, WCP itself has higher water absorption and water demand, which can adversely affect the mechanical and working properties of cementitious materials, and this effect becomes larger with increases in WCP admixture. However, WCP itself has potential activity, and the activity index of WCP can reach up to 80 after activity activation, so the WCP activated by the optimal activation method has less influence on the product properties [25].

According to the physical properties, chemical composition, microstructure, and activity index, it is clear that WCP has a small particle size and large specific surface area, which is conducive to the filling effect of "micro aggregate". In addition to the high water absorption capacity, the activity index of activated WCP is also high. These factors make WCP potentially active and indicate the feasibility of WCP as an auxiliary cementing material. However, the main chemical composition of WCP is complex and highly variable due to its different sources, which is one of the reasons why WCP is not efficiently applied.

**Table 1.** Main chemical compositions of WCP.

| No. | Source | SiO₂ (%) | CaO (%) | Fe₂O₃ (%) | Al₂O₃ (%) | MgO (%) | SO₃ (%) | K₂O (%) | Na₂O (%) |
|---|---|---|---|---|---|---|---|---|---|
| 1 | [16] | 27.80 | 29.10 | 2.73 | 6.70 | 4.49 | 1.11 | - | 1.21 |
| 2 | [2] | 20.50 | 62.2 | 3.48 | 5.32 | 2.73 | - | 1.02 | 0.72 |
| 3 | [24] | 31.00 | 54.90 | 1.90 | 3.40 | 5.60 | 0.98 | 1.31 | 0.79 |
| 4 | [26] | 49.06 | 32.49 | 3.45 | 8.29 | 2.16 | 1.66 | 1.57 | 0.67 |
| 5 | [27] | 39.85 | 30.84 | 2.03 | 6.34 | 2.19 | 1.93 | 1.76 | 1.13 |
| 6 | [28] | 51.80 | 22.80 | 5.60 | 13.60 | 1.40 | 0.80 | 2.60 | - |
| 7 | [29] | 51.80 | 22.81 | 5.59 | 13.58 | 1.39 | 0.79 | 2.64 | 0.69 |
| 8 | [30] | 49.97 | 18.65 | 2.30 | 8.89 | 1.37 | 2.53 | 3.35 | 0.80 |
| 9 | [31] | 29.74 | 47.22 | 3.77 | 3.17 | 1.24 | 1.83 | 0.54 | - |
| 10 | [32] | 58.55 | 11.82 | 4.64 | 10.35 | 1.52 | 0.44 | - | 0.78 |

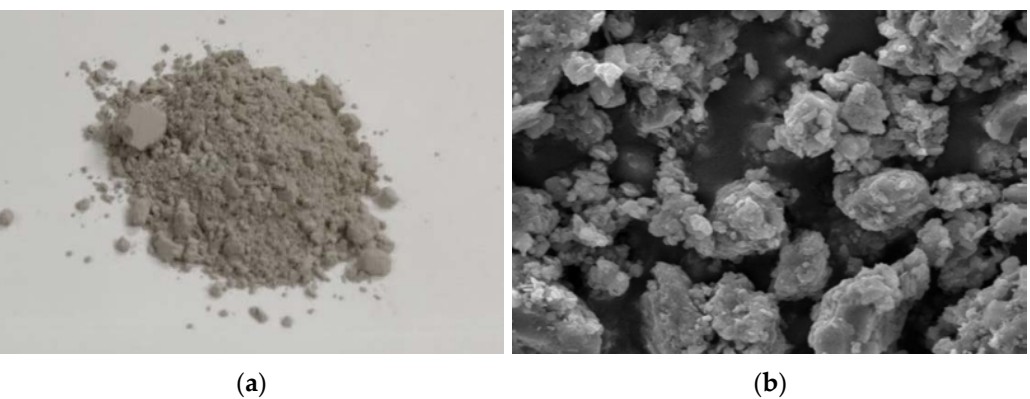

(**a**)  (**b**)

**Figure 1.** (**a**) WCP [30]. (**b**) SEM diagram of WCP [11].

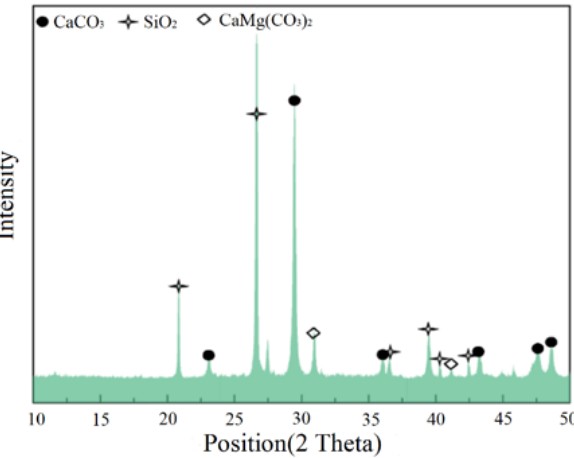

**Figure 2.** XRD diagram of WCP Reprinted with permission from Ref. [24]. 2023, Huixia Wu.

## 3. Activation of Waste Concrete Powder

The main components of WCP are $SiO_2$, CaO, and $Al_2O_3$ are shown in Table 1. The percentage contents of these three components are high and can directly participate in hydration, which can improve the early mechanical properties of the cementitious material [33]. However, there are some components of WCP, such as calcium alumina, calcium hydroxide, and calcium silicic aluminate, that will have different percentage contents due to the different sources of WCP, and although they cannot directly participate in the hydration reaction, certain methods can be adopted that can stimulate their potential activity and make them become cementitious materials with high hydration properties, either by reducing the huge energy and environmental load brought about by cement production

or by creating waste concrete. The complete and efficient recycling of waste concrete is of great significance [34].

At present, the activity excitation of WCP has become a research hotspot for scholars at home and abroad, and the activity index of WCP can best reflect the mechanical properties and activation effect of the products. According to the determination method of the WCP activity index in JG/T573-2020 *Recycled Micronized Powder for Concrete and Mortar*, WCP is used to replace 30% of the cement in the preparation of cement, and after 28 days of curing, the compressive strength is measured and compared with the mortar without WCP, that is, the activity index (SAI) of WCP is obtained, and an SAI > 60 is Grade I recycled micronized powder, and 60 < SAI < 70 is Class II recycled micronized powder. In this paper, through the study of a large amount of literature, the three activation methods of mechanical activation, thermal activation, and chemical activation are compared and analyzed using the activity index of WCP as the standard, aiming to provide a reference for the efficient utilization of regenerated micronized powder and subsequent research.

In this study, first, we used Citespace software to summarize the research on WCP activation modes in recent years. In Figure 3, We can clearly see that the keyword co-occurrence network above is clustered into color regions, each corresponding to a label. The order is from 0 to 4. The smaller the number, the more keywords are contained in the clusters, and each cluster is composed of multiple closely related words. In Figure 3, it can be seen that the most researched topic in chemical activation by domestic and foreign scholars is the activation of WCP activity by alkali, followed by thermal activation, and the study of WCP activation cannot be separated from the mechanical properties of WCP products. Then, Citespace software was used to count the keyword relevance of WCP research in the last 14 years in relation to time, and the thicker the curve, the tighter the relevance. In Figure 4, it can be seen that the research on WCP before 2014 was mainly focused on the mechanical properties of the products, and after 2014, it was mainly focused on the activation and working properties of WCP. The longest duration of research on chemical activation has been from 2009 to now.

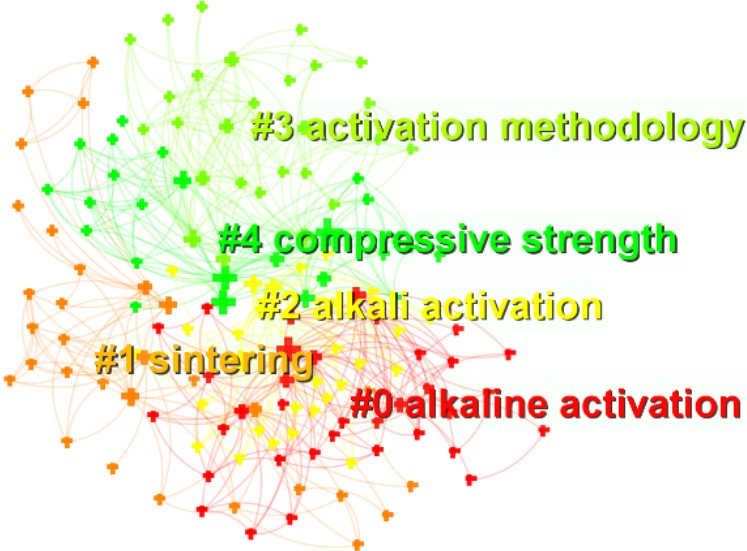

**Figure 3.** Keyword clustering of WCP activation modes.

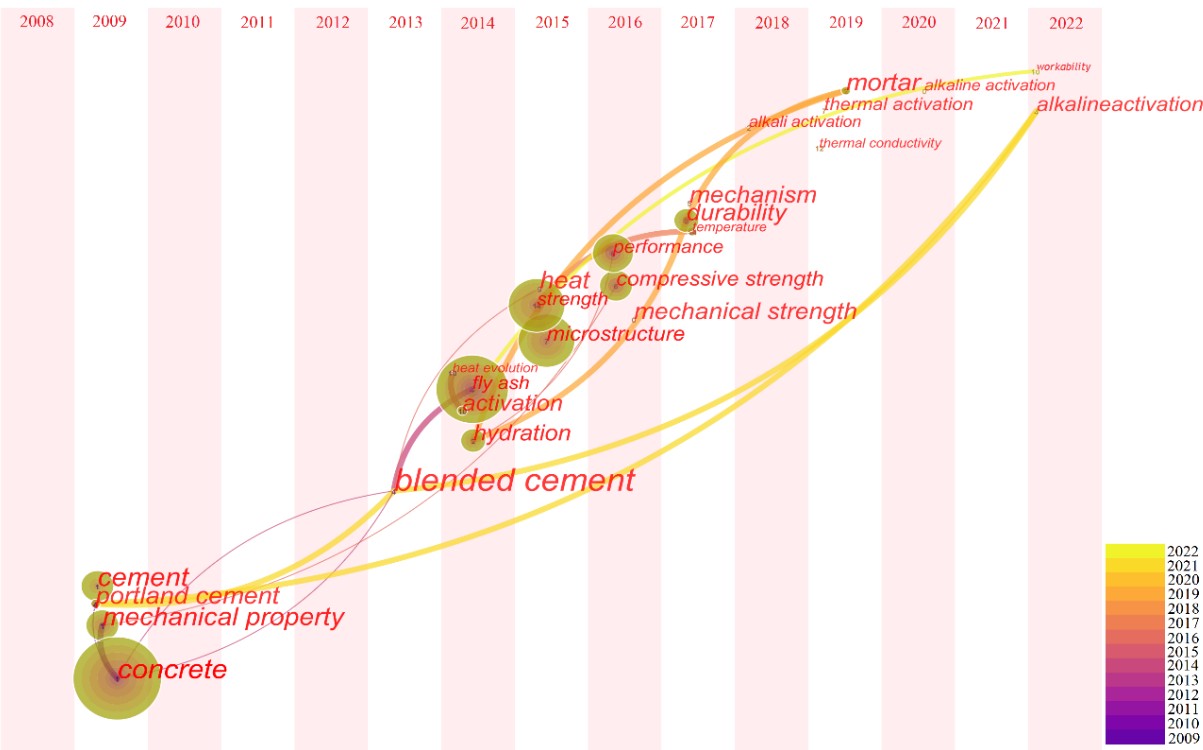

**Figure 4.** WCP keywords time zone diagram.

### 3.1. Mechanical Activation

Mechanical activation refers to the use of mechanical grinding to activate WCP. Mechanical grinding can reduce WCP's particle diameter and increase its specific surface area, and the stable crystalline $\alpha$-SiO$_2$ in WCP is transformed into the more stable $\beta$-SiO$_2$, and finally, into amorphous SiO$_2$ by mechanical force [35]. Meanwhile, under the action of mechanical grinding, the original regular hydration products Ca(OH)$_2$ and C-S-H crystals in the cement slurry continuously collide and eventually become amorphous [36]. However, WCP should not be ball milled too finely, otherwise, it will lead to agglomeration of the material and reduce the strength of the specimen.

Yang [13] found that the WCP activity was best after 1 h of ball milling, but its growth was not obvious with increases in ball milling time after 60 min, because the WCP particles were too fine, which might reduce the compactness of the void filling in the cementitious sand and decrease the strength of the specimens. A similar theory was derived from the study of Lou [37]. Li [38] found that controlling the WCP particle size around 75 μm resulted in better activity. Sun [39] considered that WCP obtained by ball milling for 150 min had a passage rate of 78% for a 75 μm sieve, which met the requirements for the particle size of regenerated powder in China, but the ball milling time was too long, which led to a decrease in the efficiency of powder grinding and lower economic efficiency. Ling [2] found that once the ball milling time of WCP exceeded 60 min, its grinding efficiency was greatly reduced. As shown in Figure 5, in 30–60 min, the particle size of WCP changed significantly, and the efficiency of grinding after 60 min was not ideal. The activity index of WCP mechanically activated for 60 min was 67.68 and the activity of WCP after 75 min ball milling was 69.22, see Figure 5. Considering the energy consumption, a ball milling time of 60 min was optimal.

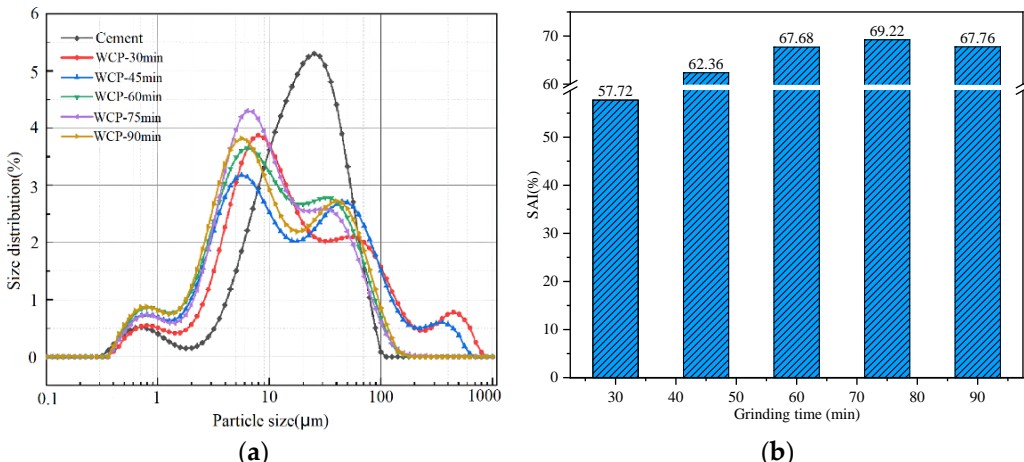

**Figure 5.** (**a**) Mechanical activation of WCP particle size distribution. (**b**) Mechanical activation of WCP particle size distribution. Reprinted with permission from Ref. [2]. 2023, Dongsheng Zhang.

In summary, the activity of WCP can be effectively improved by mechanical grinding, but too long a ball milling time will lead to a decrease in the WCP activity instead. Considering the mechanical activation effect and energy consumption, it is suggested that the optimal grinding time for WCP is 60 min. Since mechanical activation can increase the specific surface area of WCP and make its reaction more adequate, mechanical activation can be considered a pretreatment step of chemical activation and thermal activation so that the latter two activation effects are better.

### 3.2. Chemical Activation

Chemical activation is the process of improving the hydration and hardening ability of cementitious materials by using organic or inorganic chemical activators to produce a cementitious system with high strength and high water demand to improve activity [40]. Chemical activation methods include mono-alkali activation, mono-salt activation, and compound activation, and the commonly used strong alkalis include CaO, NaOH, and $Ca(OH)_2$. The commonly used salts include $CaSO_4$, $Na_2SO_4$, NaCl, and $CaCl_2$ [2]. Most of the current studies have focused on single exciter activation; however, research on complex activation is still in its infancy.

### 3.2.1. Mono-Alkali Activation

WCP activity can be effectively stimulated in an alkaline environment. On the one hand, free $Ca^{2+}$ reacts with $SiO_4^{2-}$ to form C-S-H gel, which increases the specimen's strength [41]. On the other hand, in an alkaline environment, the reactive $Al_2O_3$ and $SiO_2$ in WCP react with $OH^-$, and the Si-O and Al-O bonds are more easily broken [42], increasing the amount of generated gels, such as C-S-H and C-A-H. However, excessively strong alkali decreases the $Ca^{2+}$ concentration, the amount of $Ca(OH)_2$, and the cement hydration rate, leading to a decrease in the strength of the specimen.

CaO can provide an alkaline environment for WCP, and CaO generates a lot of heat when dissolved in water, which acts as a thermal activator, and thus, improves the early strength of WCP products. CaO can also provide the necessary $Ca^{2+}$ for the gelling products of C-S-H, C-A-H, C-A-S-H, and AFt [43]. Liu [43] discovered that combining CaO and gypsum produces mine-filled gelling materials with high early strengths. Zhang [19] found that the flexural and compressive strengths of WCP-prepared mortar reached the maximum when the CaO content was 3% and the WCP activity index reached the highest value of 78.31, see Figure 6. $Ca(OH)_2$ generated by the reaction of CaO and water then reacted with $SiO_2$ and $Al_2O_3$ to form C-S-H and C-A-H, respectively, with the following reactions:

$$CaO + H_2O \rightarrow Ca(OH)_2 \downarrow \tag{1}$$

$$mCa(OH)_2 + SiO_2 + nH_2O \rightarrow mCaO \bullet SiO_2 \bullet nH_2O \tag{2}$$

$$mCa(OH)_2 + Al_2O_3 + nH_2O \rightarrow mCaO \bullet Al_2O_3 \bullet nH_2O \tag{3}$$

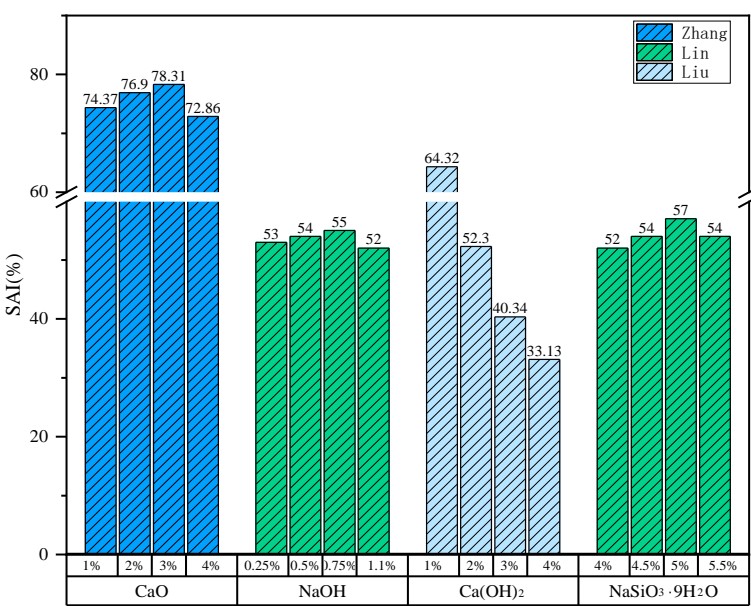

**Figure 6.** CaO, NaOH, Ca(OH)$_2$, and Na$_2$SiO$_3 \cdot$9H$_2$O effects on WCP activity index. adapted with permission from Refs. [2,15,44] 2023, Dongsheng Zhang, Zhongcai Lin, and Yin Liu.

NaOH ionizes OH$^-$ in water, OH$^-$ reacts with Ca$^{2+}$, the free Ca$^{2+}$ reacts with SiO$_4{}^{2-}$ to form C-S-H gels, and Na$^+$ can promote silicate network depolymerization [45,46]. Dong [47] found that the strength of WCP specimens after 48 h of autoclave maintenance was gradually increased with increasing NaOH dosing when the NaOH dosing was within the range of 0% to 4.8%. Xu [48] found that for ordinary WCP specimens, the specimen strength increased and then decreased as the NaOH doping amount increased, and when the reference amount was 0.75%, the strength reached the maximum value and the WCP activity index reached the highest value of 55, as shown in Figure 6. The reaction equation in an alkaline environment is as follows:

$$2OH^- + Ca^{2+} \rightarrow Ca(OH)_2 \downarrow \tag{4}$$

$$mCa^{2+} + SiO_4{}^{2-} + nH_2O \rightarrow mCaO \bullet SiO_2 \bullet nH_2O \tag{5}$$

Ca(OH)$_2$ not only provides an alkaline environment for the reactive silica and aluminous components in waste concrete raw materials to react and produce geopolymer gels, but it also provides the required Ca$^{2+}$. That is, after Ca(OH)$_2$ alkali activation, waste concrete can produce gels such as C-S-H and C-A-H [47,49]. Some scholars [32,41] have found that some Ca(OH)$_2$ could not significantly improve the compressive and flexural strength of the specimens at the early stage (3 and 7 days). However, when the age of maintenance was extended to 28 d, the strength of the recycled mortar was significantly improved. Other scholars [42,43] found that when Ca(OH)$_2$ was dosed at 1%, the compressive strength at the ages of 3 d and 7 d decreased, and the strength at the age of 28 d increased because the Ca(OH)$_2$ could not dissolve rapidly to cause a hydration reaction in the early stage but could react with 3CaO-Al$_2$O$_3$ in cement in the later stage to form a large amount of C$_3$AH$_6$, increasing the strength of mortar specimens. Mixing 1% Ca(OH)$_2$ exciter effectively excited the activity of WCP, and its activity index reached the highest value of 64.32; see Figure 6. The reactions of active SiO$_2$, Al$_2$O$_3$, and Ca(OH)$_2$ with water are shown in Equations (2) and (3).

The Na$_2$SiO$_3$·9H$_2$O activation mechanism provides the same alkaline environment as NaOH to promote Si-O and Al-O bond breaking [47], and SiO$_3^{2-}$ is able to promote the generation of gelling substances such as C-S-H and C-A-H [42]. Rong [50] found that the 3 d and 28 d flexural and compressive strengths of specimens were increased when Na$_2$SiO$_3$·9H$_2$O with a modulus of 1.4 was incorporated into WCP. Xu [43] concluded that with an increase in NaSiO$_3$·9H$_2$O exciter doping, the strength of the specimens first increased and then decreased, and the highest activity index of the WCP was 57 at 5% doping; see Figure 6. The following is the reaction of SiO$_3^{2-}$ with Ca$^{2+}$ to form the C-S-H reaction:

$$SiO_3^{2-} + Ca^{2+} + H_2O \rightarrow CaO \bullet SiO_2 \bullet H_2O \tag{6}$$

### 3.2.2. Mono-Salt Activation

Mono-salt activation is mainly divided into sulfate and chloride salts. Sulfate salts can provide SO$_4^{2-}$ and promote the production of Ettringite (AFt). Cl$^-$ in chloride salts can react with activated Al$_2$O$_3$ to produce hydrated calcium chloroaluminate and increase the osmotic pressure of the system. Chloride salts can react with Ca(OH)$_2$ to produce a gelling component and improve the specimen's strength [48]. Zhang [2] found that the incorporation of CaSO$_4$ could not improve the early strength of WCP mortar, but it could improve its later compressive strength. The strength reached a maximum when the CaSO$_4$ admixture was 1%, and the WCP activity index reached a maximum value of 76.44, as shown in Figure 7.

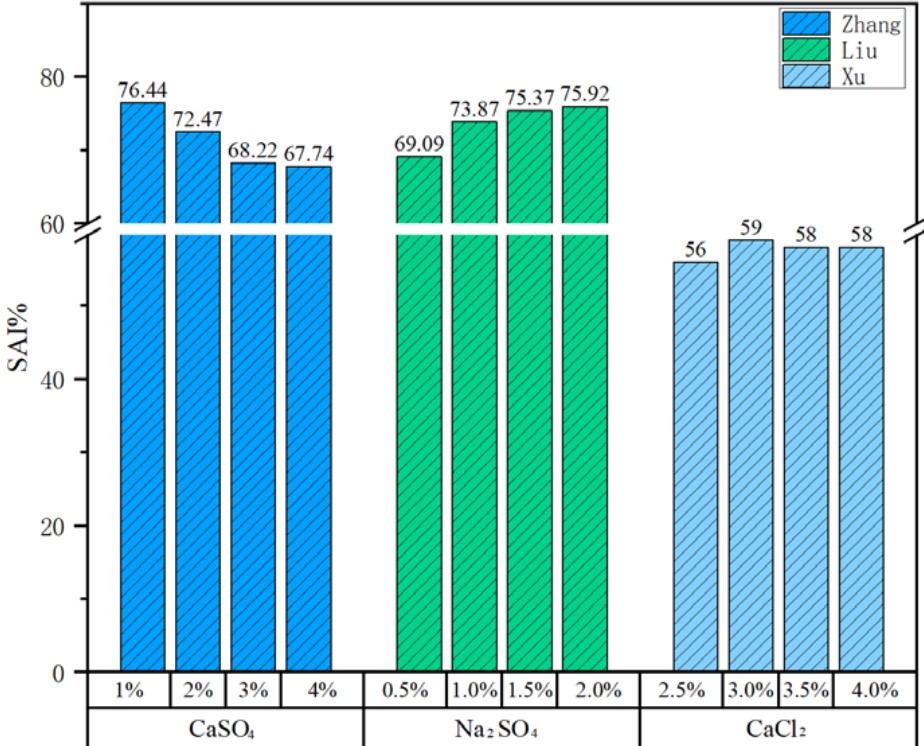

**Figure 7.** Effects of CaSO$_4$, Na$_2$SO$_4$, and CaCl$_2$ on the activity index of WCP adapted with permission from Refs. [2,44,48]. 2023, Dongsheng, Zhang, Rulin Xu and Yin Liu.

Na$_2$SO$_4$ also has a better activation effect because Na$_2$SO$_4$ can react with Ca(OH)$_2$ to generate highly dispersed CaSO$_4$ and also AFt. Some scholars [15,49] have found that the compressive strength of the specimens increased with an increase in Na$_2$SO$_4$ doping, and the effect was better at 2% doping, which was due to the reaction of the Na$_2$SO$_4$ with Ca(OH)$_2$ to generate CaSO$_4$, which accelerated the rate of hydration reaction of C$_3$S, generating more C-S-H gels and improving the strength of the specimens. The WCP activity index reached a maximum value of 75.92; see Figure 7. Through experiments, some

researchers [51] came to the conclusion that the CaSO$_4$ produced by the reaction of Na$_2$SO$_4$ with Ca(OH)$_2$ was also capable of reacting with the reactive Al$_2$O$_3$ in the system to produce needle-like AFt, which improved the specimen's strength. The sulfate reaction equation is as follows:

$$AlO_2^- + Ca^{2+} + OH^- + SO_4^{2-} \rightarrow 3CaO \bullet Al_2O_3 \bullet 3CaSO_4 \bullet 32H_2O \tag{7}$$

It was shown in [34] that chlorine salts can both change the osmotic pressure of the system and participate in the reaction to generate gelling substances and improve the strength of the specimens. Some researchers [48,52,53] have found that when CaCl$_2$ was added to WCP, the compressive strength and flexural strength of the specimens increased and then decreased with an increase in CaCl$_2$ dosing. The strength reached the maximum value when the dosing was 3.5%, and the WCP activity index reached the maximum value of 59; see Figure 7. Because the addition of NaCl or CaCl$_2$ can generate hydrated calcium chloroaluminate and increase the osmotic pressure of the system, and because CaCl$_2$ can also react with Ca(OH)$_2$ to generate a cementitious component and increase the strength of the specimen, NaCl should not be used in reinforced concrete because Na$^+$ and Cl$^-$ will be introduced at the same time. The chloride–salt reaction is as follows:

$$Ca^{2+} + Al_2O_3 + Cl^- + OH^- \rightarrow 3CaO \bullet Al_2O_3 \bullet CaCl_2 \bullet 10H_2O \tag{8}$$

### 3.2.3. Salt–Alkali Complex Activation

Compared to the activity of alkali-excited or salt-excited modified WCP, salt–alkali complex activation combines the advantages of the former two excitants, which can further promote the hydration reaction, produce more hydration products, form a denser microstructure, and improve the strength of WCP concrete and mortar [54]. Zhang [2] studied the effects of different salt-base combinations and their different ratios on the WCP activity index when the total admixture was 3%. The WCP activity index was 76.96 when the ratio of CaO to Na$_2$SO$_4$ was 1:2; the WCP activity index was 73.82 when the ratio of CaO to CaSO$_4$ was 1:2; the WCP activity index was 72.17 when the ratio of Ca(OH)$_2$ to Na$_2$SO$_4$ was 1:1; and the WCP activity index was 80.27 when the ratio of Ca(OH)$_2$ and CaSO4 was 1:1; see Figure 8. Xu [48] also carried out a similar study, and the WCP activity index was 69 when the ratio of NaOH to CaSO$_4$ was 1:1; the WCP activity index was 62 when the ratio of Na$_2$SiO$_3$·9H$_2$O to CaSO$_4$ was 1:2. When the ratio of Na$_2$SiO$_3$·9H$_2$O to CaCl$_2$ was 1:2, the WCP activity index was 58; see Figure 8.

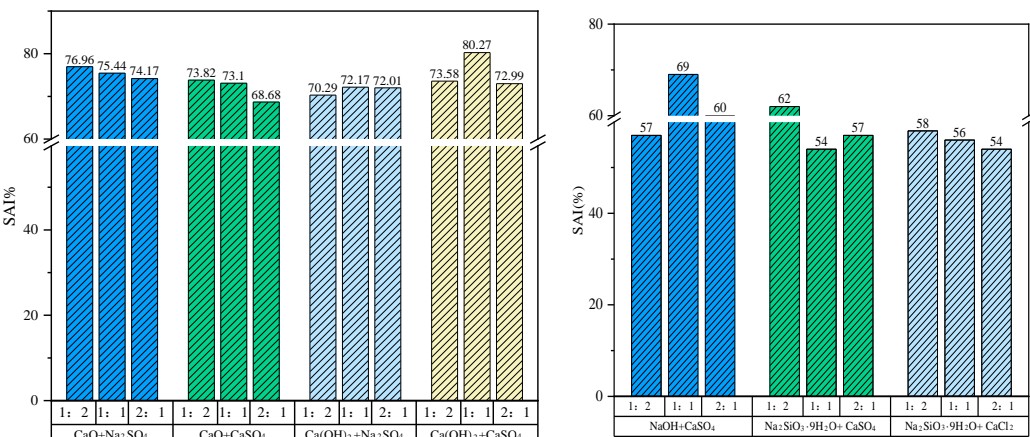

**Figure 8.** Salt–alkali complex activation (3% doping). Reprinted with permission from Refs. [2,48] 2023, Dongsheng Zhang, Rulin Xu.

In summary, the activity of WCP can be effectively improved by adding a mono-alkali, mono-salt, or salt–alkali mixture. However, the activation effects of different activators were very different, and between the single admixture activators, the mono-alkali had the best activation effect with CaO, and the WCP activity index after activation was 78.31. The activation effect of $CaSO_4$ was the best with mono-salt, and the WCP activity index after activation was 76.44. Among the compound activations, $Ca(OH)_2$ plus $CaSO_4$ had the best activation effect, and the WCP activity index after activation was 80.27. Table 2 summarizes and analyzes the influence of the above three activators with the best activation effect on the mechanical properties of the mortar made by WCP. A comprehensive analysis determined that the activation effect of 3% CaO on WCP was better. Because mechanical activation increases the specific surface area of WCP and causes it to react more fully, it can be considered a pretreatment step of chemical activation to improve the chemical activation effect.

**Table 2.** Effects of CaO, $CaSO_4$, $Ca(OH)_2+CaSO_4$ on the mechanical properties of recycled mortar (dosing of 3%).

| Activator | Mechanical Properties of Mortar | Source |
|---|---|---|
| CaO | Compressive strength: Pre-strength and post-strength are improved; Flexural strength: Strength decreases in the early stage, strength increases in the later stage. | [15,27,33] |
| $CaSO_4$ | Compressive strength: Late strength increase; Flexural strength: Strength decreases in the early stage, strength increases in the later stage. | [15,24,39] |
| $Ca(OH)_2 + CaSO_4$ | Compressive strength: Strength decreases in the first period, strength increases in the later period; Flexural strength: Strength decreases in the early stage, strength increases in the later stage. | [15,31,39] |

### 3.3. Thermal Activation

Thermal activation refers to the activation of the potential activity of mineral admixtures by increasing the temperature. Since the reaction rate of cementitious materials is closely related to temperature, the particle size of WCP decreases after high-temperature treatment, so this method can maximize the potential activity of WCP [11,55]. Since WCP is mainly derived from cement slurry in waste concrete, its composition includes C-S-H gel, $Ca(OH)_2$, AFt, and other phases, of which fully hydrated C-S-H gel accounts for about 70% of the total volume, $Ca(OH)_2$ for about 20%, AFt and other substances for about 7%, and unhydrated particles and impurities for about 3% [56].

This paper summarizes and analyzes the effect of temperature on the main components of WCP, C-S-H gel, $Ca(OH)_2$, and Aft, and explores the activation characteristics of thermally activated WCP. High-temperature treatment of C-S-H gel resulted in high hydration activity [57]. Peng [58] calcined waste mortar at 400–800 °C. C-S-H was found to start decomposing at 560 °C, and the decomposition rate increased significantly with an increasing temperature above 600 °C, which was the main reason for the mass loss of waste concrete at temperatures above 600 °C. C-S-H decomposition products included -$C_2S$ and $C_3S$. Rodriguez [59] investigated C-S-H at high temperatures by preparing C-S-H with calcium–silica ratios ranging from 0.75 to 1.5 and temperatures ranging from >1000 °C. The results revealed that the calcium–silica ratio affected the temperature required for the structural transition, -$C_2S$ was produced by the dehydration of C-S-H with a calcium–silica ratio greater than 1.0, and the different calcium–silica ratios directly affected the temperature of the C-S. According to Shumei Yu [60], as the calcium–silica ratio increased from 0.8 to 1.7, the temperature of C-S-H decomposition decreased from 650–900 °C to 400 °C, i.e., the higher the calcium–silica ratio, the lower the decomposition temperature. At high temperatures, the decomposition of $Ca(OH)_2$ into CaO, the destruction of $Ca(OH)_2$

layered macro-crystals, and the increase in contact points can improve the overall activity of WCP [61]. Some scholars [55,62,63] have found that $Ca(OH)_2$ gradually decomposes at 300–800 °C, and at about 600 °C, $Ca(OH)_2$ decomposes basically completely. AFt at high temperatures can be dehydrated to form $Al_2O_3$ colloids and CaO with certain gelling abilities, which in turn improves the activity of WCP [34]. Other scholars [55,58,59] have found that AFt was gradually dehydrated from 75 to 300 °C, and at about 150 °C, the AFt was basically completely dehydrated.

In summary, the thermal decomposition temperature of C-S-H and $Ca(OH)_2$ is around 600 °C, and that of AFt is around 150 °C; see Table 3. The activity of WCP after high-temperature treatment is significantly increased, and the highest activity of WCP was 80.47 when the thermal activation temperature reached 800 °C; see Figure 9. On the one hand, this is because, after the high-temperature treatment, the bound water within this WCP disappears. On the other hand, the decomposition of C-S-H, $Ca(OH)_2$, and AFt was complete at 800 °C, which produced a large amount of active substances. These products can be hydrated when they meet water and have a certain gelling ability, but due to the different sources of WCP, its composition varies greatly, and the activity of WCP after thermal activation also varies.

**Table 3.** Effect of heat treatment on the main components of WCP.

| Hydration Products | Decomposition Temperature | Decomposition Products | Source |
|---|---|---|---|
| C-S-H | 600 °C | $\beta$-$C_2S$, $C_3S$ | [41,55,64] |
| $Ca(OH)_2$ | 600 °C | CaO | [41,55,64] |
| AFt | 150 °C | $Al_2O_3$, CaO | [51,55,56] |

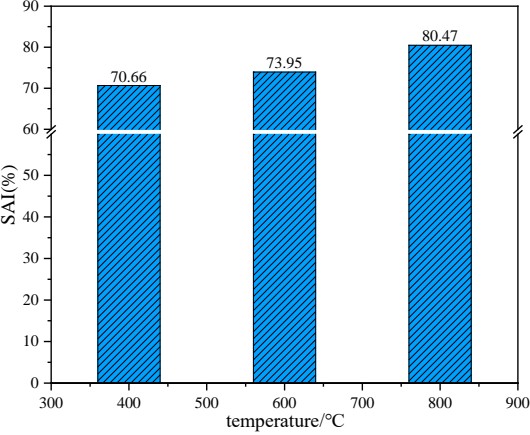

**Figure 9.** Effects of thermal activation on WCP activity index. Reprinted with permission from Ref. [19]. 2023, Jianzhuang Xiao.

### 3.4. WCP Activation Method Advantages, Disadvantages, and Feasibility Analysis

From the analysis, organization, and summary of the existing literature, it is clear that the academic and industrial communities at home and abroad have realized the importance of WCP as an alternative gelling material. Due to the low activity of WCP, more studies on the active excitation of WCP have been conducted in recent years. Table 4 summarizes the advantages and disadvantages of different active excitation methods and their feasibility, aiming to provide a reference for the efficient application of WCP. Table 4 shows that although the effect of mechanical excitation is not the best, it is the most widely used excitation method due to its simple operation and low production cost. However, if an efficient and inexpensive chemical excitation agent can be found, then chemical excitation is the best choice.

Table 4. Analysis of the advantages, disadvantages, and feasibility of the excitation method.

| Activation Method | Advantages | Disadvantages | Feasibility Analysis | Source |
|---|---|---|---|---|
| Mechanical activation | Mechanical excitation can increase the particle fineness of WCP, which can effectively excite its activity, and the method is simple to operate, widely used in ball mills, and has low activation costs. | The long ball milling time will make the WCP particles too fine, and its activity will be reduced instead, so the method has limited improvement of the WCP activity index; the highest is only 69.22. | The method requires ball milling with a ball mill, but the power consumption is large, about 8 degrees/hour per ton of WCP processed, but the method is cheap, so its use is the most widespread. | [37,48,53] |
| Chemical activation | Chemical activation can provide an alkaline environment or reactive ions, and this method excites significantly, with a maximum activity index of 80.2 after WCP activation. | The method will lead to a large difference in the activation effect due to different sources of WCP, and different chemical excitants have different optimal dosing, which has certain inconvenience in practical application, and the excitant is more expensive. | The method requires the addition of an exciter, which is more expensive, so the method is currently limited to the experimental stage, and an efficient and inexpensive activator needs to be found. | [12,46,61] |
| Thermal activation | Thermal activation gives WCP the ability to rehydrate and gel by changing the compositional structure of the original material. This method has the best excitation effect and is simple to operate, with the highest activity index of 80.47 after WCP activation. | Due to the wide range of WCP sources, the process, temperature, and heating rate during thermal activation have some uncertainty regarding their activation effects on WCP from different sources, and the method has high energy consumption. | The method uses a heating furnace for 600–800 °C high-temperature heating; although the power consumption is high, the method has the best activation effect, and the heating temperature is much lower than the temperature of calcined cement, so the method has a certain market. | [60,63,65] |

## 4. Reuse of Waste Concrete Powder in Mortar

WCP has a similar chemical composition to cement with potential volcanic ash activity, which makes WCP-prepared mortars workable, and most of the current research by scholars has focused on the application of WCP as an admixture in mortars [66]. Yong [32] showed that the fluidity and mechanical strength of mortars using WCP as an admixture decreased, and WCP was considered a non-reactive powder. Bordy [67] et al. confirmed that the compressive strength of mortars containing 24% active residual anhydrous clinker in net cement slurry and 24% net cement slurry instead of cement was reduced; however, the compressive strength of mortars with 20% cement slurry replacement could reach 83% of the benchmark specimens. Zhu [68] suggested that WCP containing unhydrated cement particles could partially replace silica fume or cement as a concrete active powder in concrete. Although WCP contains cement paste, the unhydrated cement particles in the cement paste are related to the water–cement ratio (w/c) of the concrete; the lower the w/c, the higher the possibility of the presence of unhydrated cement particles in the concrete [69].

In this study, first, we used Citespace keyword clustering to count the number of studies on the reuse of WCP in recent years. In Figure 10, We can clearly see that the keyword co-occurrence network above is clustered into color regions, each corresponding to a label. The order is from 0 to 4. The smaller the number, the more keywords are contained in the clusters, and each cluster is composed of multiple closely related words. We then ranked the research hotspots. In Figure 10, it can be seen that the subject of the most research by domestic and foreign scholars has been the preparation of recycled mortar using WCP, followed by the preparation of mixed mortar by compounding WCP with

other external admixtures. Then, the Citespace keyword network was used to analyze the correlations among the performance studies of WCP products in recent years. As can be seen in Figure 10, the research on WCP products is mainly divided into mechanical properties research and workability research, so this paper summarizes and analyzes the mechanical properties and workability of WCP mortar products, aiming to provide reference for the efficient utilization of WCP.

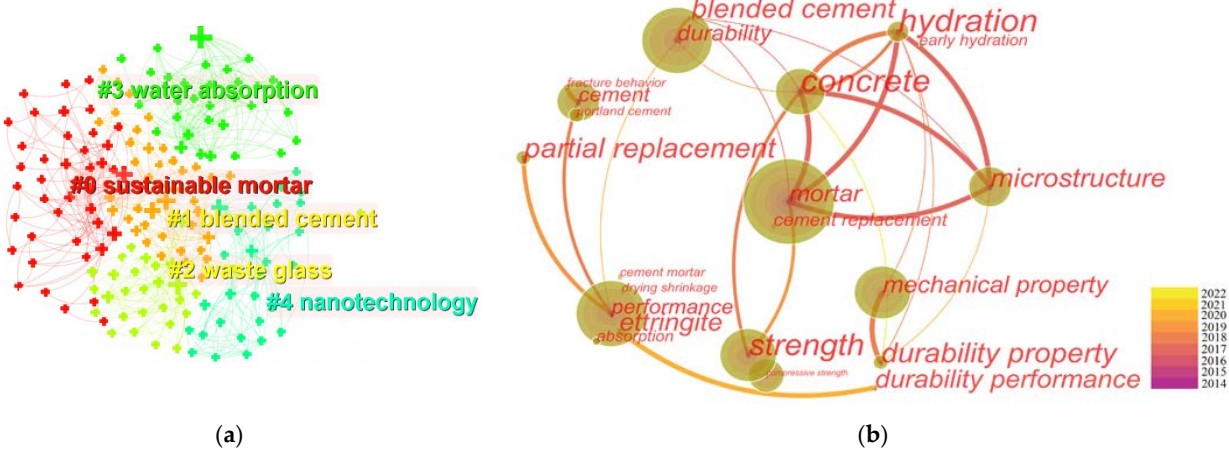

(**a**)    (**b**)

**Figure 10.** (**a**) Keyword clustering for WCP reuse. (**b**) WCP keyword co-occurrence network.

### 4.1. Mechanical Properties

#### 4.1.1. Compressive Strength

Compressive strength is an important index reflecting the mechanical properties of cementitious materials, which vary under different variables. Duan [56], Wu [24], and Zhang [65] found through their experiments that the compressive strength of cement mortar showed a decreasing trend with the addition of WCP when the water–cement ratio was certain. The compressive strength of the mortar met the requirements with small doses of admixture, reaching 70–80% of the compressive strength of ordinary mortar, while the compressive strength of the cement mortar decreased significantly with a larger admixture of WCP. As can be seen in Figure 11, the compressive strength of cement mortar decreased significantly with the increase in the WCP admixture, which was due to the irregular shape of WCP particles, the increase in the water requirement at the same fluidity, and the decrease in the cement content in the mortar, resulting in a decrease in the compressive strength [70].

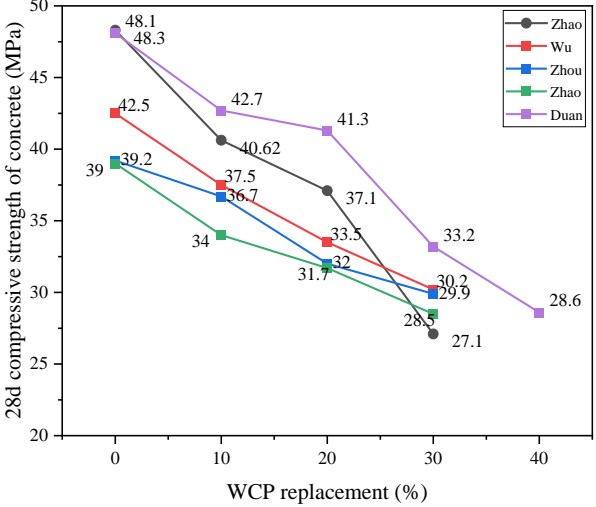

**Figure 11.** Compressive strength of mortars with different WCP substitution rates [24,56,71–73].

### 4.1.2. Flexural Strength

The research on WCP by domestic and foreign scholars has mainly focused on the effects of WCP admixture on the mechanical properties of cement mortar. Regarding the effects of WCP admixture on the flexural strength of cement mortar, the conclusions reached by domestic and foreign scholars [24,56,73–75] basically coincide. Figure 12 shows a line graph based on the test data of domestic and foreign researchers. In the figure, it can be seen that the flexural strength of cement mortar tends to decrease with increases in WCP admixture, and it decreases significantly when the WCP admixture exceeds 20%. This is because WCP contains a large amount of fine powder (particles below 10 μm in size), which fill the pores in the cement paste, and a small amount of admixture has little effect on the strength of the mortar. However, because unactivated WCP has low activity of its own, its main enhancement of the cement mortar performance comes from its own micro-aggregate filling effect [73]. Since the strength of cement mortar mainly comes from the formation of hard cementite and dense gel tissue with cement hydration, when the WCP dosing is too high, the active material involved in the hydration reaction in the whole cementitious system is significantly reduced, and the micro-aggregate filling effect of WCP is not enough to offset the effect of the reduction of cement clinker on the mechanical properties, resulting in a significant decrease in the degree of hydration of the recycled mortar compared to the ordinary mortar, and therefore, the cement mortar loses a significant amount of strength [25,76].

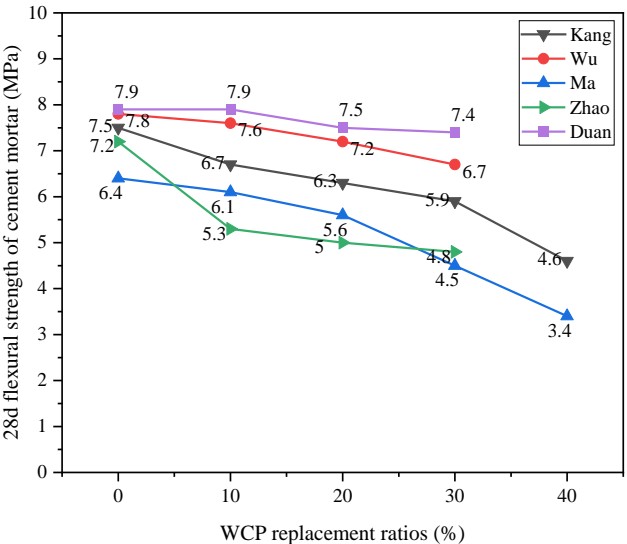

**Figure 12.** Flexural strength of mortars with different WCP substitution rates [20,53,70–72].

### 4.1.3. Splitting Tensile Strength

The splitting tensile strength is a measure of the tensile properties of concrete. The incorporation of WCP adversely affects the splitting tensile strength of concrete; the higher the incorporation amount, the more obvious it is. Fan [23], Peng [77], and Fan [78] have found that the overall splitting tensile strength of recycled concrete tends to decrease with an increase in the WCP substitution rate. However, the amount of WCP dosed below 10% produced fewer negative effects or even weak benefits, and the strength decreased faster above 10%. The appropriate mix ratio had a positive effect on the splitting tensile strength of concrete; see Figure 13. For example, Liu et al. [79] found that the 28 d splitting tensile strength could be improved when the substitution rate was 10% with the addition of composite powder.

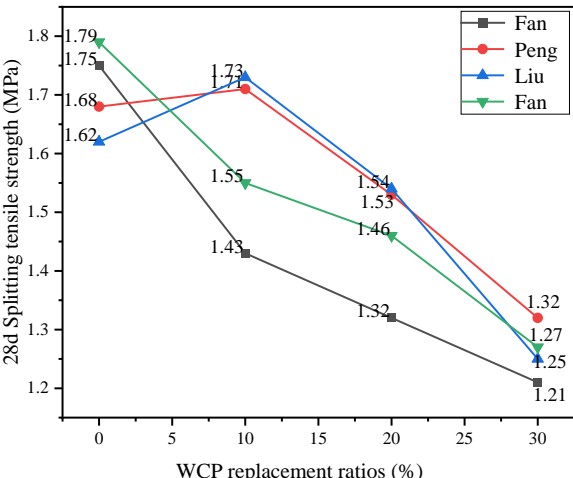

**Figure 13.** Splitting tensile strength of concrete at different WCP substitution rates [23,77–79].

In summary, the admixture of WCP harms the compressive strength, flexural strength, and splitting tensile strength of concrete. Because of the different sources of WCP and the large number of inert components in untreated WCP, the admixture of WCP should not be higher than 20%, considering the strength requirements of cementitious materials. To ensure that the mechanical properties of the products will not be degraded while increasing the WCP admixture, the results of this study indicate that WCP prepared from waste concrete with a small water–cement ratio can be selected and treated with a combination of three excitation methods—mechanical excitation, chemical excitation, and thermal excitation—to maximize the WCP activity.

*4.2. Working Performance*

4.2.1. Setting Time

The time required for a mortar to gradually lose its fluidity and plasticity with the hydration reaction and to have a certain strength is the setting time [78]. The setting time is one of the important indicators used to judge the early performance of cementitious materials. Since WCP has less active material, the hydration reaction of regenerated mortar produces less cementitious material, and the setting time is prolonged [79,80]. Chen [25], Larsen [81], and Liu [82] have reached similar conclusions. However, some scholars have come to the opposite conclusion. Zhu [83] and Yang [31] have found that the setting time of mortar was shortened with increasing WCP admixture. Figure 14 summarizes the experimental data from several research scholars. Although the setting time of mortar has varied, it was discovered that they all met the specifications for the mortar setting time. The incorporation of WCP will reduce the participation of cement particles in the hydration reaction and reduce the production of cementitious materials. Moreover, it will also reduce the release of the heat of hydration and slow the setting time of mortar [74]. However, WCP acts as a crystal nucleus and promotes the generation of hydration products [84]. Therefore, the effects of WCP incorporation on the setting time of mortar will vary depending on the source of WCP.

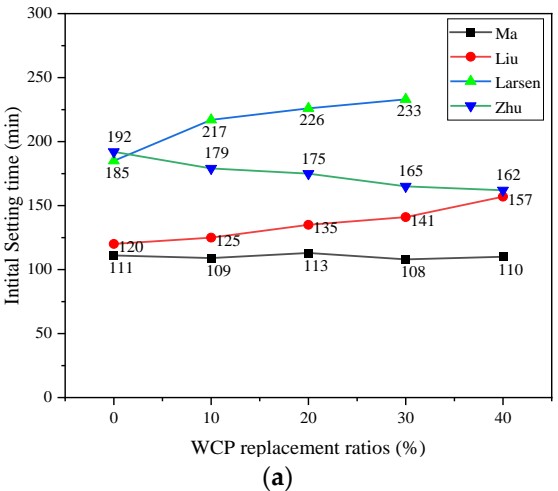 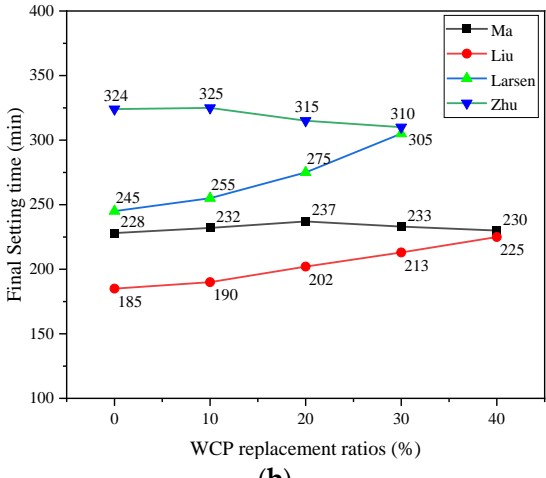

**Figure 14.** Setting time of mortars with different WCP substitution rates [74,81–83]. (**a**) Initial setting time; (**b**) final setting time.

### 4.2.2. Workability

Workability is an important indicator reflecting the working performance of cement–cement mortar. Zhu [83], Chen [85], and Liu [82] found that the incorporation of WCP will reduce the workability of mortar. On the one hand, because WCP particles are smaller than cement particles and have a larger specific surface area, the water content between the pores of cement particles will be relatively reduced under the same water–cement ratio, resulting in lower mortar workability [86]. The small size of WCP can play a filling role between cement particles, which increases the resistance of net slurry flow, thus leading to the reduction of mortar workability. On the other hand, the WCP surface is rough and irregular, which increases the resistance of mortar particles to move with each other [82,87,88]. Figure 15 summarizes the effect of WCP incorporation on cement mortar workability. It can be seen in the figure that the admixture of WCP adversely affected the mortar's fluidity. However, when the WCP admixture was less than 20%, the reduction in mortar fluidity was less than 10% in all cases. Although the admixture of WCP adversely affected the mortar workability, the small amount of admixture had a limited effect on the mortar workability.

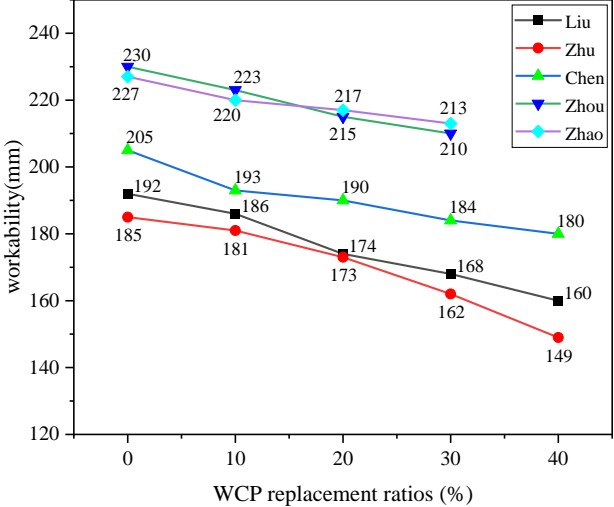

**Figure 15.** Workability of mortar with different WCP substitution rates [72,73,82,83,85].

### 4.2.3. Dry Shrinkage

Drying shrinkage is one of the factors reflecting the working performance of mortar, and there have not been many studies on the effect of WCP on the drying shrinkage of mortar. Wu [89] found that the effect of WCP on the drying shrinkage of mortar was minimal compared to other regenerated micronized powders when the dosing amounts were all 30%. Zhang [90] and Ji [80] found that the drying shrinkage of mortar became smaller as the dosing amount of WCP increased; see Figure 16. Therefore, the incorporation of WCP can improve the drying shrinkage of mortar. On the one hand, because there are more inert substances in WCP, it is not conducive to the early hydration reaction of mortar, which reduces the consumption of water, makes its internal environment milder, and reduces the drying shrinkage [91]. On the other hand, due to the small particle size of WCP, it plays a micro-aggregate filling role, reducing the internal pore space of the material and enhancing its drying shrinkage resistance [92].

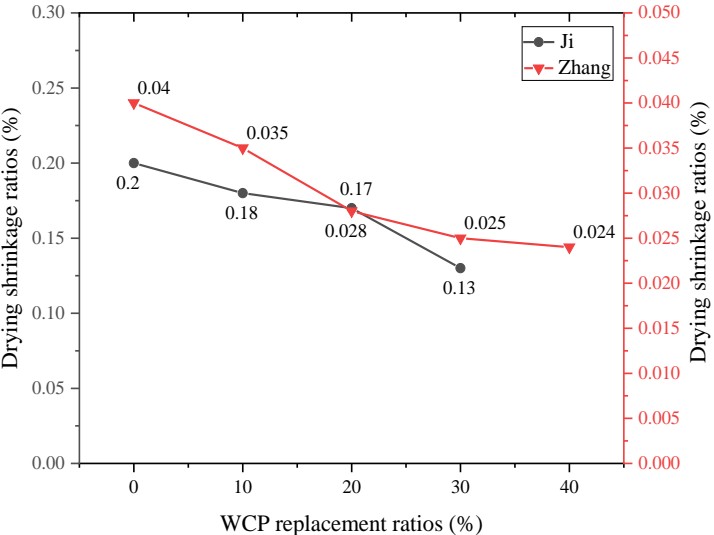

**Figure 16.** Drying shrinkage of cement mortar with different WCP substitution rates [80,90].

In summary, the effects of WCP on the working properties of mortar are as follows: (1) The setting time of regenerated mortar is mainly influenced by the grade and source of WCP. The influence of WCP of different grades and sources on the setting time of mortar varies, but most of the prepared regenerated mortar setting times meet the specification requirements. (2) The fluidity of recycled mortar is mainly influenced by the particle size and source of WCP. When the water–cement ratio is the same, the smaller the WCP particle size, the smaller the fluidity of the recycled mortar, which is because the small particle size can play a filling role, and the rough surface of WCP increases the resistance of the mortar flow. Therefore, WCP has a negative impact on the fluidity of mortar, and the amount of mixture is most effective when it is under 20%. (3) The drying shrinkage of recycled mortar is mainly affected by the particle size and source of WCP. WCP with a smaller particle size will play a filling role in the mortar and enhance the compactness of the mortar, thus reducing the drying shrinkage of the mortar. When WCP comes from waste concrete with a larger water–cement ratio, the more inert internal components it has, and the greater the inhibitory effect it has on mortar drying shrinkage. Therefore, WCP has a favorable effect on the drying shrinkage of mortar, and the larger the amount of WCP, the smaller the drying shrinkage of the mortar.

## 5. Conclusions

In the process of recycling waste concrete, WCP is a solid waste material with great potential application value. Many scholars have conducted a lot of research on the activation of WCP, and the main conclusions are as follows:

(1) Mechanical activation increases the fineness of WCP particles to effectively activate its activity. This method is simple and low-cost, but a ball milling time that is too long will lead to particles that are too fine. In addition, the activity is reduced, and the method has a limited effect on WCP activity stimulation.

(2) Chemical activation mainly provides an alkaline environment or reactive ions to stimulate the potential activity of WCP. Chemical activation has a significant excitation effect, but due to the strong fluctuation of the composition of WCP, different chemical excitants have different optimal doses. The practical application of chemical activation has a certain inconvenience, and it is more expensive.

(3) Thermal activation mainly changes the composition of the original material structure so that the WCP has the ability for rehydration gelation. However, due to the wide range of WCP sources, the chemical composition of WCPs from different sources varies somewhat, leading to some uncertainties in their thermal activation effects. Despite the significant excitation effect of this method, its energy consumption is high.

(4) Most researchers have used WCP to manufacture mortar and concrete. However, the cementitious properties of WCP after active excitation are also inferior to those of cement. Therefore, the performance of its products is not satisfactory. Activated WCP can be compounded with other active powders.

Due to the different sources of WCP, resulting in its complex composition and various activation methods, there are still some shortcomings in the research on WCP, and further research is needed to promote its application in practical engineering:

(1) Because the different sources of WCP have a greater impact on the performance of WCP products, to make WCP widely used, it is necessary to promote the establishment of a waste concrete inspection and classification system.

(2) More research on single exciters—and less research on compound exciters—is needed to find efficient and inexpensive exciters in order to make chemical excitation in actual engineering applications more widely used.

(3) The activation effect of thermal activation is obvious and this method is simple to operate. Thermal activation can be studied in depth to determine the optimal thermal activation temperature for different sources of WCP and to establish a reliable thermal activation standard.

(4) At present, most of the studies have focused on the mechanical properties and working performance of WCP products, but the durability of WCP products has been less studied and should be studied in depth.

**Author Contributions:** C.B. was mainly responsible for determining the general direction of writing and improving the paper in the final process. B.T. was responsible for writing the whole article, reviewing all the documents, and for the first translation. Q.W. was mainly responsible for downloading and filing the documents and also checking and correcting the writing of the paper. Y.Q. was responsible for the table layouts and translation of the paper. Y.S. mainly reviewed the translation of the paper and guided its logical framework. L.Y. and Q.Y. were responsible for the finalization and revision of the paper and for the collection of excellent literature in the field of waste concrete powder. All authors have read and agreed to the published version of the manuscript.

**Funding:** This research was funded by the project of the Natural Science Foundation of Chongqing municipality (cstc2021jcyj-msxmX0444), the project of the Chongqing Construction Science and Technology Plan (2021 No. 1-6), and the project of the Chongqing Bureau of Human Resources and Social Security (cx2020008). This study was also supported by the open fund of Chongqing Key Laboratory of Energy Engineering Mechanics & Disaster Prevention and Reduction (EEMDPM2021103), and the fund of the State Key Laboratory of Bridge Engineering Structural Dynamics, Key Laboratory of Bridge Earthquake Resistance Technology, Ministry of Communications, PRC. The project was also supported by the scientific and technological research program of the Chongqing Municipal Education Commission (KJZD-K202201503).

**Institutional Review Board Statement:** Not applicable.

**Informed Consent Statement:** Not applicable.

**Data Availability Statement:** Not applicable.

**Acknowledgments:** Thanks to all those who helped write this review.

**Conflicts of Interest:** The authors declare no conflict of interest.

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
