# Peer review of "Activation Method and Reuse of Waste Concrete Powder—A Review"

_sustainability, doi:10.3390/su15065451_

Round 1

Reviewer 1 Report

Manuscript Number: Sustainability-2234874

Title: Active excitation and reuse of waste concrete powder - A review

This review paper summarizes the findings in the field of utilizing the Waste Concrete Powder for construction industry. In spite of the attempt made by the authors, the paper requires a major improvement to match the journal standards. Moreover, the author just summaries the research work of others. In addition, the number of articles used to justify the existing findings will not be sufficient for a standard review article. Further improvement is also required to improve the discussions based on the parameters. Hence, the reviewer suggests a major revision before making a final recommendation for the manuscript. The following comments are to be considered for further improvement.

Comments and Suggestions for the Manuscript:

1.    Page 1 Abstract “WCP” - Expanded form of the abbreviation should be provided at the first instance.

2.    Page 1 Introduction – provide current scenario on the quantum of construction waste generate globally with proper literature support.

3.    Page 2 “In this paper, we firstly ……….. Product properties of WCP” sentence has to revised for better understand in the view of readers.

4.    Page 2 Section 2 “The XRD patterns of WCP are shown in Figure 2. It can be seen from the figure that the main mineral components of WCP are SiO2, CaCO3, CaMg(CO3)2, etc.,” Provide a literature support.

5.    Page 2 Section 2 “However, due to the potential activity of WCP itself, the activated WCP can reach up to 80% of the activity of ordinary cement, so the activated WCP with the best activation method has less influence on the product 's performance. all of which have potential activation properties”

Provide justification on How the activated WCP have potential activity increased upto 80%?

6.    Page 3 Section 3: “WCP contains a large number of potentially active silica-aluminous components” Justification needed in terms of chemical reaction stating the degree of reactivity on adding or replacing the WCP with aggregate or binder.

7.    Page 3 “ The mortar was prepared by using WCP instead of 30% cement, and  after 28 days of maintenance, the compressive strength after 28 days was measured and compared with the mortar without WCP, that is, the activity index (SAI) of WCP was obtained, and SAI > 60 was the grade regenerated micronized powder, and 60 < SAI < 70 was the grade regenerated micronized powder. Provide literature support.

8.    Page 7: [47]. “Zhang [19] found that the incorporation of CaSO4 did not improve the early strength of WCP mortar. However, with the increase in curing age, the compressive strength of the recycled mortar first increased and then decreased with the increase in CaSO4 admixture.” The sentence is unclear and please elaborate for the benefit of the readers.

9.    Page 13 “This may be due to the fact that WCP contains a …….. will affect the mortar strength.”

Author has to justify whether hydration of mortar is affected   

10. Conclusion has been written in more of a generic sense. Hence, the author has been suggested to revise in consideration with the context of parameters considered.

Author Response

Response to Reviewer 1 Comments

Point 1: Page 1 Abstract “WCP” - Expanded form of the abbreviation should be provided at the first instance.

Reply: Thank you for pointing out the relevant missing expressions in the paper, and we have made the relevant improvements.

Revision: We have added an extended form of the first "WCP" abbreviation to the abstract.

Point 2: Page 1 Introduction – provide current scenario on the quantum of construction waste generate globally with proper literature support.

Reply: Thank you for your advice, we have provided the current status of the total global construction waste generation in the introduction on page 1, supported by appropriate literature.

Revision: With the accelerated urbanization, the annual production of construction waste exceeds 3.5 billion tons worldwide, which poses a great challenge to landfill disposal of construction waste as well as environmental protection [1,2].

Point 3: Page 2 “In this paper, we firstly …… Product properties of WCP” sentence has to revised for better understand in the view of readers.

Reply: Thank you for your suggestion, we have made the modification, so that readers can better understand.

Revision: We found that this sentence duplicates the meaning of the last paragraph of the introduction, so we revised the sentence and incorporated it into the last paragraph for better understanding by the reader.

Point 4: Page 2 Section 2 “The XRD patterns of WCP are shown in Figure 2. It can be seen from the figure that the main mineral components of WCP are SiO2, CaCO3, CaMg(CO3)2, etc.,” Provide a literature support.

Reply: Thank you for the reminder, we have added a reference for this sentence.

Revision: The XRD patterns of WCP are shown in Figure 2. It can be seen from the figure that the main mineral components of WCP are SiO2, CaCO3, CaMg(CO3)2, etc., all of which have potential activation properties [24].

Point 5: Page 2 Section 2 “However, due to the potential activity of WCP itself, the activated WCP can reach up to 80% of the activity of ordinary cement, so the activated WCP with the best activation method has less influence on the product 's performance. all of which have potential activation properties”

Provide justification on How the activated WCP have potential activity increased upto 80%?

Reply: According to the determination method of WCP activity index in JG/T573-2020 "Reclaimed Micro Powder for Concrete and Mortar", using WCP instead of 30% cement to prepare colloidal sand and curing for 28 days, the ratio of the compressive strength of the colloidal sand to that of ordinary colloidal sand is the activity index (SAI) of WCP. The highest activity index of the literature in the chart below has reached 80, that is, the compressive strength of WCP colloidal sand has reached 80% of the compressive strength of ordinary colloidal sand. We have modified it for the reader's convenience.

Revision: However, WCP itself is potentially active, and the activity index of WCP can reach up to 80 after active activation, so the WCP activated by the optimal activation method has less influence on the product performance [25].

Point 6: Page 3 Section 3: “WCP contains a large number of potentially active silica-aluminous components” Justification needed in terms of chemical reaction stating the degree of reactivity on adding or replacing the WCP with aggregate or binder.

Reply: Thank you for the reminder that we did not express this sentence clearly enough, so we have revised it.

Revision: The main components of WCP are SiO2, CaO and Al2O3 are shown in Table 1. The content of these three components is high and can directly participate in hydration, which can improve the early mechanical properties of the cementitious material [35]. However, there are also some parts within WCP such as calcium alumina, calcium hydroxide and calcium silicic aluminate, which will have different contents due to the different sources of WCP, and although they cannot directly participate in the hydration reaction, certain methods can be taken that can stimulate their potential activity and make them become cementitious materials with high hydration characteristics.

Point 7: Page 3 “ The mortar was prepared by using WCP instead of 30% cement, and  after 28 days of maintenance, the compressive strength after 28 days was measured and compared with the mortar without WCP, that is, the activity index (SAI) of WCP was obtained, and SAI > 60 was the grade Ⅰ regenerated micronized powder, and 60<SAI<70 was the grade Ⅱ regenerated micronized powder. Provide literature support.

Reply: Thanks for the heads up, we've cited the source for this quote.

Revision: According to the determination method of WCP activity index in JG/T573-2020 "Recycled Micronized Powder for Concrete and Mortar", WCP is used to replace 30% cement in the preparation of mastic sand, and after 28 days of curing, the compressive strength after 28 days is measured and compared with the mortar without WCP, that is, the activity index (SAI) of WCP is obtained, SAI>60 is Grade I recycled micronized powder, 60<SAI<70 is Grade II recycled micronized powder.

Point 8: “Zhang [19] found that the incorporation of CaSO4 did not improve the early strength of WCP mortar. However, with the increase in curing age, the compressive strength of the recycled mortar first increased and then decreased with the increase in CaSO4 admixture.” The sentence is unclear and please elaborate for the benefit of the readers.

Reply: Thank you for your suggestion, we have revised the sentence for better understanding by readers.

Revision: Zhang [2] found that the incorporation of CaSO4 could not improve the early strength of WCP mortar, but it could improve its later compressive strength..

Point 9: Page 13 “This may be due to the fact that WCP contains a ……. will affect the mortar strength.”

Author has to justify whether hydration of mortar is affected.

Reply: A small amount of WCP has little effect on the mortar strength and can play a micro-aggregate filling effect. However, the active substance in WCP is less than that in cement. When the content of WCP is too high, the active substance involved in hydration reaction in the whole gelling system will be significantly reduced, and the micro-aggregate filling effect of WCP is not enough to offset the effect of the reduction of cement clinker on the mechanical properties, which will have an adverse effect on the hydration of mortar. Therefore, when the content of WCP is too large, the strength of cement mortar will be significantly reduced. We have added notes and cited new literature in the text for better understanding by the reader.

Revision: This is because WCP contains a large amount of fines (particles with a particle size below 10 μm), which fill the pores in the cement paste, and a small amount of admixture has little effect on the mortar strength. However, because the unactivated WCP has a low activity of its own, its main enhancement of the cement mortar performance comes from its own microaggregate filling effect [78]. Since the strength of cement mortar mainly comes from the formation of hard cementite and dense gel tissue with cement hydration, when the WCP dosing is too high, the active material involved in the hydration reaction in the whole cementitious system is significantly reduced, and the micro-aggregate filling effect of WCP is not enough to offset the effect of the reduction of cement clinker on the mechanical properties, resulting in a significant decrease in the degree of hydration of the recycled mortar compared with the ordinary mortar, and therefore the cement mortar significant loss of strength [25,81]..

Point 10: Conclusion has been written in more of a generic sense. Hence, the author has been suggested to revise in consideration with the context of parameters considered.

Reply: Thank you for your valuable comments. In the conclusion section, we summarized and analyzed the existing problems of the WCP research Institute and the focus of future WCP research.

Revision: In the process of recycling waste concrete, WCP is a solid waste material with great potential application value, many scholars have conducted a lot of research on the activation of WCP, and the main conclusions are as follows:

(1) mechanical activation by increasing the fineness of WCP particles to effectively activate its activity, the method is simple and low cost, but the ball milling time is too long will lead to too fine particles, but the activity is reduced, and the method has limited effect on WCP activity stimulation.

(2) Chemical activation is mainly to provide alkaline environment or reactive ions to stimulate the potential activity of WCP. Chemical activation has a significant excitation effect, but due to the strong fluctuation of the composition of WCP, different chemical excitants have different optimal dosing, the practical application has certain inconvenience, and the chemical activation is more expensive.

(3) Thermal activation mainly through changing the composition of the original material structure, so that WCP has the ability to rehydration gelation. However, due to the wide range of WCP sources, the process, temperature and heating rate of thermal activation have some uncertainty on the activation effect of WCP from different sources. Despite the significant excitation effect of this method, its energy consumption is high.

(4) Most researchers use WCP to make mortar and concrete. However, the cementitious properties of WCP after active excitation are also inferior to those of cement. Therefore, the performance of its products is not satisfactory. The activated WCP can be compounded with other active powders.

Due to the different sources of WCP, resulting in its complex composition and various activation methods, there are still some shortcomings in the research of WCP, and further research is needed to promote its application in practical engineering:

(1) Due to the different sources of WCP have a greater impact on the performance of WCP products, to make WCP widely used, it is necessary to promote the establishment of a waste concrete inspection and classification system.

(2) More research on single exciter, but less research on compound exciter, need to find efficient and inexpensive exciter, in order to make chemical excitation in the actual engineering more widely used.

(3) The excitation effect of thermal activation is significant and simple to operate, should be studied in depth from different strengths of waste concrete WCP thermal excitation of the optimal temperature, the establishment of sound related thermal activation standards.

(4) At present, most of the studies focus on the mechanical properties and working performance of WCP products, but the durability of WCP products is less studied, which can be studied in depth.

Reviewer 2 Report

This study presents a review of many literatures related to reuse of waste concrete powder (WCP) in mortar and concrete applications. The available activation methods of the WCP were described, compared, and analyzed together with the activity index of the reference WCP. These included mechanical, chemical and thermal activations utilized to achiave an efficient WCP. The paper is interesting for the readers and informative. The methodology was organized in a logical way. However, most of the sentences were long and they need to be shorter. In addition, duplicated names were given for Sections 4.1 and 4.2 . If these comments are taken into consideration, I recommend accepting this paper to be published in the Sustainability Journal.

Author Response

Response to Reviewer 2 Comments

Overview

This study presents a review of many literatures related to reuse of waste concrete powder (WCP) in mortar and concrete applications. The available activation methods of the WCP were described, compared, and analyzed together with the activity index of the reference WCP. These included mechanical, chemical and thermal activations utilized to achiave an efficient WCP. The paper is interesting for the readers and informative. The methodology was organized in a logical way. However, most of the sentences were long and they need to be shorter. In addition, duplicated names were given for Sections 4.1 and 4.2. If these comments are taken into consideration, I recommend accepting this paper to be published in the Sustainability Journal.

Point 1: Most of the sentences were long and they need to be shorter.

Reply: We double-checked the manuscript and the article does have the problem of too many long sentences, and we have tried to make some changes. By the way, this manuscript submitted to MDPI for English editing has been edited by a professional company. The final result can be seen in the revised article.

Point 2: Duplicated names were given for Sections 4.1 and 4.2.

Reply: Thank you for the heads up, we have found the error and made the changes.

Revision: 4.1 Mechanical properties, 4.2 Working properties.

Reviewer 3 Report

The author reviews the active excitation and reuse of waste concrete powder, The research is interesting and valuable, but the content depth and analysis are lacking. It is recommended to reconsider after an overhaul.

For the abstract, What are the conclusions drawn after the review of this work, what are the deficiencies, and what needs to be improved?

The research on WCP is still in its initial stages. It is wrong; please revise it.

The greenhouse gas emissions from construction waste and carbon dioxide emissions from building materials need to be introduced.

In the introduction part, it is necessary to introduce the research work of relevant researchers on reducing greenhouse, such as https://doi.org/10.1016/j.jobe.2022.104880.

https://doi.org/10.1016/j.conbuildmat.2022.126921.

The logic of the introduction part is confusing and needs to be revised.

The physical properties of waste concrete need to be summarized

Summarize the advantages and disadvantages of several activation methods in detail

Fig.7, add some references to support your analysis.

Is there any difference between 4.1 and 4.2? Please revise it.

For 4.1, mechanical properties only include compressive and flexural strength; please add other strengths.

In the conclusion part, summarize the problems in this research, the focus of the next research, and the problems that may be encountered in the end.

Author Response

Response to Reviewer 3 Comments

Overview

The author reviews the active excitation and reuse of waste concrete powder, The research is interesting and valuable, but the content depth and analysis are lacking. It is recommended to reconsider after an overhaul.

  1. For the abstract, what are the conclusions drawn after the review of this work, what are the deficiencies, and what needs to be improved?
  2. The research on WCP is still in its initial stages. It is wrong; please revise it.
  3. The greenhouse gas emissions from construction waste and carbon dioxide emissions from building materials need to be introduced.

In the introduction part, it is necessary to introduce the research work of relevant researchers on reducing greenhouse, such as https://doi.org/10.1016/j.jobe.2022.104880.

https://doi.org/10.1016/j.conbuildmat.2022.126921.

  1. The logic of the introduction part is confusing and needs to be revised.
  2. The physical properties of waste concrete need to be summarized.
  3. Summarize the advantages and disadvantages of several activation methods in detail.
  4. Fig.7, add some references to support your analysis.
  5. Is there any difference between 4.1 and 4.2? Please revise it.
  6. For 4.1, mechanical properties only include compressive and flexural strength; please add other strengths.
  7. In the conclusion part, summarize the problems in this research, the focus of the next research, and the problems that may be encountered in the end.

Point 1: For the abstract, what are the conclusions drawn after the review of this work, what are the deficiencies, and what needs to be improved?

Reply: Thank you for your valuable comments. we have made changes to the abstract.

Revision: In this paper, by reviewing relevant literature at home and abroad in recent years, we firstly used Citespace software to visualize and analyze the research on the reuse and activation methods of WCP in recent years, then explained the characteristics of WCP and the effects of different activation methods on WCP activity index, and summarized the mechanical properties and working properties of WCP mortar products, and finally analyzed the best activation method of WCP and the best mixing amount of WCP for mortar preparation, in addition, some problems in the current research were also prospected.

Point 2: The research on WCP is still in its initial stages. It is wrong; please revise it.

Reply: Thank you for pointing out our mistake, we have made changes to it.

Revision: Currently, most of the research on waste concrete is focused on the preparation of recycled aggregates, but there is less research on waste concrete powder (WCP).

Point 3: The greenhouse gas emissions from construction waste and carbon dioxide emissions from building materials need to be introduced.

In the introduction part, it is necessary to introduce the research work of relevant researchers on reducing greenhouse, such as https://doi.org/10.1016/j.jobe.2022.104880.

https://doi.org/10.1016/j.conbuildmat.2022.126921.

Reply: Your suggestion is very valuable, and we have included in the introduction the CO2 emissions of building materials and present the research work of relevant researchers in reducing greenhouse.

Revision: Cement-based materials, as commonly used cementitious materials, are widely used in the construction industry due to the easy availability and low cost of raw materials. Currently, the world produces 4.2 billion tons of silicate cement annually, which is the largest man-made material in the world, and 7% of the annual global CO2 emissions come from cement production [5,6]. excessive CO2 emissions have caused many global problems such as global warming, sea level rise, and glacier melting [7]. It is imperative to save energy and reduce CO2 emissions [8]. Cui [7] used sulfo-aluminate cement instead of normal silicate cement to prepare low-carbon eco-ultra-high performance concrete. And WCP, as a by-product of waste concrete disposal process, has a similar chemical composition to cement, and the use of WCP to replace some cement in mortar and concrete has been shown to be environmentally friendly and effective [9].

Point 4: The logic of the introduction part is confusing and needs to be revised.

Reply: Thank you for your suggestion, we have revised the introductory section.

Revision: We include a paragraph in the introduction on CO2 emissions from cement production and analyze the need for WCP preparation. Finally, we removed repetitive meaningful sentences and summarized in the last paragraph of the introduction.

Point 5: The physical properties of waste concrete need to be summarized.

Reply: Thank you for your valuable suggestions, we have summarized the physical properties of WCP.

Revision: Compared with silicate cement, WCP itself has higher water absorption and greater water demand, which will adversely affect the mechanical and workability of cementitious materials, and this effect will become greater with the increase of WCP admixture. However, WCP itself is potentially active, and the activity index of WCP can reach up to 80 after activation, so the WCP activated by the optimal activation method has less influence on the product properties [25].

According to the physical properties, chemical composition, microstructure and activity index, WCP has small particle size and large specific surface area, which is beneficial to the filling effect of "microaggregate". In addition to its high water absorption capacity, the activated WCP has a high activity index. These factors make WCP potentially active and indicate the feasibility of WCP as an auxiliary cementing material. However, the main chemical composition of WCP is complex and highly variable due to its different sources, which is one of the reasons why WCP is not efficiently applied.

Point 6: Summarize the advantages and disadvantages of several activation methods in detail.

Reply: Thank you for your suggestion, we have summarized the advantages and disadvantages of the various WCP activation methods and the feasibility analysis in Table 5.

Revision: Table 5 Analysis of the advantages, disadvantages and feasibility of the excitation method

Activation method

Advantages

Disadvantages

Feasibility Analysis

Source

Mechanical activation

Mechanical excitation can increase the particle fineness of WCP, which can effectively excite its activity, and the method is simple to operate, widely used in ball mills, and has low activation cost.

The long ball milling time will make the WCP particles too fine, and its activity will be reduced instead, so the method has limited improvement of WCP activity index, the highest is only 69.22.

The method requires ball milling with a ball mill, but the power consumption is large, about 8 degrees per hour for each ball mill 1 ton of WCP, but the method is cheap, so its use is the most widespread.

[37],[47],, [72]

Chemical activation

Chemical activation can provide an alkaline environment or reactive ions, and this method excites significantly, with a maximum activity index of 80.2 after WCP activation.

The method will lead to a large difference in activation effect due to different sources of WCP, and different chemical excitants have different optimal dosing, which has certain inconvenience in practical application, and the excitant is more expensive.

The method requires the addition of exciter, for example, 3% doping, each activation of 1 ton of WCP consumes about 3 kg of exciter, expensive, and so the method is currently limited to the experimental stage, the need to find a highly efficient and inexpensive activator.

[12],[46],[77]

Thermal activation

Thermal activation gives WCP the ability to rehydrate and gel by changing the composition structure of the original material. This method has the best excitation effect and is simple to operate, with the highest activity index of 80.47 after WCP activation.

Due to the wide range of WCP sources, the process, temperature and heating rate during thermal activation have some uncertainty on the activation effect of WCP from different sources, and the method has high energy consumption.

The method needs to use the heating furnace for 600-800 ℃ high temperature heating, although the power consumption is high, but the method has the best activation effect, and the heating temperature is much lower than the temperature of calcined cement, so the method has a certain market.

[60],[63],[75]

Point 7: Fig.7, add some references to support your analysis.

Reply: Thanks for your advice, we have added references and data for Figure 7.

Revision:

图7 CaSO4、Na2SO4和CaCl2对WCP的活性指数影响[2,47,48]

Point 8: Is there any difference between 4.1 and 4.2? Please revise it.

Reply: Thanks for pointing out the error, we have corrected this error.

Revision: 4.1 Mechanical properties, 4.2 Working properties.

Point 9: For 4.1, mechanical properties only include compressive and flexural strength; please add other strengths.

Reply: We have analyzed the splitting tensile strength of concrete with different WCP substitution rates in 3.1.3.

Revision: 4.1.3 Splitting tensile strength

The splitting tensile strength is a measure of concrete tensile properties. The addition of WCP has an adverse effect on the splitting tensile strength of concrete, and the more the addition amount is, the more obvious it is. Fan[23], Peng[82] and Fan[83] et al. found that with the increase of WCP substitution rate, the split tensile strength of recycled concrete showed a decreasing trend. However, when the dosage of WCP is lower than 10%, the negative effect is small, and even the benefit is weak. Proper matching ratio has positive effect on the splitting tensile strength of concrete. For example, Liu et al. [84] found that when the substitution rate was 10%, the 28d splitting tensile strength could be improved by adding compound powder.

Figure. 18 Splitting tensile strength of concrete with different WCP substitution rates[23,82–84]

Point 10: In the conclusion part, summarize the problems in this research, the focus of the next research, and the problems that may be encountered in the end.

Reply: Thank you for your valuable comments. In the conclusion section, we summarized and analyzed the existing problems of the WCP research Institute and the focus of future WCP research.

Revision: Revision: In the process of recycling waste concrete, WCP is a solid waste material with great potential application value, many scholars have conducted a lot of research on the activation of WCP, and the main conclusions are as follows:

(1) mechanical activation by increasing the fineness of WCP particles to effectively activate its activity, the method is simple and low cost, but the ball milling time is too long will lead to too fine particles, but the activity is reduced, and the method has limited effect on WCP activity stimulation.

(2) Chemical activation is mainly to provide alkaline environment or reactive ions to stimulate the potential activity of WCP. Chemical activation has a significant excitation effect, but due to the strong fluctuation of the composition of WCP, different chemical excitants have different optimal dosing, the practical application has certain inconvenience, and the chemical activation is more expensive.

(3) Thermal activation mainly through changing the composition of the original material structure, so that WCP has the ability to rehydration gelation. However, due to the wide range of WCP sources, the process, temperature and heating rate of thermal activation have some uncertainty on the activation effect of WCP from different sources. Despite the significant excitation effect of this method, its energy consumption is high.

(4) Most researchers use WCP to make mortar and concrete. However, the cementitious properties of WCP after active excitation are also inferior to those of cement. Therefore, the performance of its products is not satisfactory. The activated WCP can be compounded with other active powders.

Due to the different sources of WCP, resulting in its complex composition and various activation methods, there are still some shortcomings in the research of WCP, and further research is needed to promote its application in practical engineering:

(1) Due to the different sources of WCP have a greater impact on the performance of WCP products, to make WCP widely used, it is necessary to promote the establishment of a waste concrete inspection and classification system.

(2) More research on single exciter, but less research on compound exciter, need to find efficient and inexpensive exciter, in order to make chemical excitation in the actual engineering more widely used.

(3) The excitation effect of thermal activation is significant and simple to operate, should be studied in depth from different strengths of waste concrete WCP thermal excitation of the optimal temperature, the establishment of sound related thermal activation standards.

(4) At present, most of the studies focus on the mechanical properties and working performance of WCP products, but the durability of WCP products is less studied, which can be studied in depth.

Reviewer 4 Report

-        The manuscript tries to come up with different beneficiation and activation methods to use WCP as cement replacement. It has made a summary of relevant activation methods, which is quite plausible. However, it lacks to compare each method and come up with the best method to beneficiate WCP. Moreover, it does not tell the economic feasibility of each method. I would suggest including the economic feasibility of the activation methods to make it a complete article.

-         What does actually excitation method mean? I think it has a deeper meaning than activation and I suggest you use activation method instead of excitation. What does active excitation mean? Also, try to rephrase the title.

-        On page 2, you have mentioned that high water absorption for WCP, what is your reference here? Cement? If so, cement also has high water absorption. I am not sure about your sentence and you may need to put a reference for this.

-        Page #3, under ‘3. Activation of WCP’, How does WCP reduce the energy of cement production when it is further activated by extra chemical/thermal or mechanical methods?

-        Under the same topic, you suddenly introduced a mortar prepared by replacing 30% cement, which mortar are you refereeing to?

-         Under 3.1 Mechanical activation, you have mentioned the term ‘volcanic ash activity’. I suppose volcanic ash is different from WCP and how do you relate it with WCP?

-        On page #6, referring to the sentence … “When the maintenance period was extended to 28 days, the strength of the regenerated mortar gradually decreased with increasing Ca(OH)2 admixture and was less than 0” … do you mean the strength goes gradually to 0 or the difference in strength goes to 0? The next statement also doesn’t sound good. Because the compressive strength at an early age increased and later at 28 days it decreased, but the reason given is for an increase in compressive strength at later ages. Please look at your wording and try to rephrase it.

-        Figure 6, what is the difference in activation using CaO and Ca(OH)2 ? in an aqueous solution CaO exists in the form of Ca(OH)2. Is CaO used in solid form?

-         Full stops missing here and there through the text. Try to carefully read the article to address that.

-         Under 4.2 Mechanical properties…’coagulation time’ is not a very common term in concrete research. Do you mean the setting time of mortar/concrete?

-        The title 4.2.2 Fluidity, can be replaced with common terms in concrete/mortar research such as consistency or workability.  

Author Response

Response to Reviewer 4 Comments

Overview

  1. The manuscript tries to come up with different beneficiation and activation methods to use WCP as cement replacement. It has made a summary of relevant activation methods, which is quite plausible. However, it lacks to compare each method and come up with the best method to beneficiate WCP. Moreover, it does not tell the economic feasibility of each method. I would suggest including the economic feasibility of the activation methods to make it a complete article.
  2. What does actually excitation method mean? I think it has a deeper meaning than activation and I suggest you use activation method instead of excitation. What does active excitation mean? Also, try to rephrase the title.
  3. On page 2, you have mentioned that high water absorption for WCP, what is your reference here? Cement? If so, cement also has high water absorption. I am not sure about your sentence and you may need to put a reference for this.
  4. Page 3, under ‘3. Activation of WCP’, How does WCP reduce the energy of cement production when it is further activated by extra chemical/thermal or mechanical methods?
  5. Under the same topic, you suddenly introduced a mortar prepared by replacing 30% cement, which mortar are you refereeing to?
  6. Under 3.1 Mechanical activation, you have mentioned the term ‘volcanic ash activity’. I suppose volcanic ash is different from WCP and how do you relate it with WCP?
  7. On page 6, referring to the sentence “When the maintenance period was extended to 28 days, the strength of the regenerated mortar gradually decreased with increasing Ca(OH)2 admixture and was less than 0” … do you mean the strength goes gradually to 0 or the difference in strength goes to 0? The next statement also doesn’t sound good. Because the compressive strength at an early age increased and later at 28 days it decreased, but the reason given is for an increase in compressive strength at later ages. Please look at your wording and try to rephrase it.
  8. Figure 6, what is the difference in activation using CaO and Ca(OH)2? in an aqueous solution CaO exists in the form of Ca(OH)2. Is CaO used in solid form?
  9. Full stops missing here and there through the text. Try to carefully read the article to address that.
  10. Under 4.2 Mechanical properties…’coagulation time’ is not a very common term in concrete research. Do you mean the setting time of mortar/concrete?
  11. The title 4.2.2 Fluidity, can be replaced with common terms in concrete/mortar research such as consistency or workability.

Point 1: The manuscript tries to come up with different beneficiation and activation methods to use WCP as cement replacement. It has made a summary of relevant activation methods, which is quite plausible. However, it lacks to compare each method and come up with the best method to beneficiate WCP. Moreover, it does not tell the economic feasibility of each method. I would suggest including the economic feasibility of the activation methods to make it a complete article.

Reply: Thanks for your valuable suggestions, we have added Table 5 in 3.4 to compare and analyze the advantages and disadvantages of various activation methods and their feasibility analysis.

Revision:

2.4 WCP activation method advantages, disadvantages and feasibility analysis

From the analysis, organization and summary of the existing literature, it is clear that the academic and industrial communities at home and abroad have realized the importance of WCP as an alternative gelling material. Due to the low activity of WCP, more studies on WCP active excitation have been conducted in recent years. Table 5 summarizes the advantages and disadvantages of different active excitation methods and their feasibility, aiming to provide a reference for the efficient application of WCP. Table 5 shows that although the effect of mechanical excitation is not the best, it is the most widely used excitation method due to its simple operation and low production cost. However, if an efficient and inexpensive chemical excitation agent can be found, then chemical excitation is the best choice.

Table 5 Analysis of the advantages, disadvantages and feasibility of the excitation method

Activation method

Advantages

Disadvantages

Feasibility Analysis

Source

Mechanical activation

Mechanical excitation can increase the particle fineness of WCP, which can effectively excite its activity, and the method is simple to operate, widely used in ball mills, and has low activation cost.

The long ball milling time will make the WCP particles too fine, and its activity will be reduced instead, so the method has limited improvement of WCP activity index, the highest is only 69.22.

The method requires ball milling with a ball mill, but the power consumption is large, about 8 degrees per hour for each ball mill 1 ton of WCP, but the method is cheap, so its use is the most widespread.

[37],[47],, [72]

Chemical activation

Chemical activation can provide an alkaline environment or reactive ions, and this method excites significantly, with a maximum activity index of 80.2 after WCP activation.

The method will lead to a large difference in activation effect due to different sources of WCP, and different chemical excitants have different optimal dosing, which has certain inconvenience in practical application, and the excitant is more expensive.

The method requires the addition of exciter, for example, 3% doping, each activation of 1 ton of WCP consumes about 3 kg of exciter, expensive, and so the method is currently limited to the experimental stage, the need to find a highly efficient and inexpensive activator.

[12],[46],[77]

Thermal activation

Thermal activation gives WCP the ability to rehydrate and gel by changing the composition structure of the original material. This method has the best excitation effect and is simple to operate, with the highest activity index of 80.47 after WCP activation.

Due to the wide range of WCP sources, the process, temperature and heating rate during thermal activation have some uncertainty on the activation effect of WCP from different sources, and the method has high energy consumption.

The method needs to use the heating furnace for 600-800 ℃ high temperature heating, although the power consumption is high, but the method has the best activation effect, and the heating temperature is much lower than the temperature of calcined cement, so the method has a certain market.

[60],[63],[75]

Point 2: What does actually excitation method mean? I think it has a deeper meaning than activation and I suggest you use activation method instead of excitation. What does active excitation mean? Also, try to rephrase the title.

Reply: Thank you for your valuable advice, WCP can improve its activity index by different activation methods, so activation can indeed be a better alternative to excitation, which we have changed and modified the title.

Revision: Activation method and reuse of waste concrete powder - A review

Point 3: On page 2, you have mentioned that high water absorption for WCP, what is your reference here? Cement? If so, cement also has high water absorption. I am not sure about your sentence and you may need to put a reference for this.

Reply: Thank you for your suggestion, the high water absorption reference of WCP is cement, because there are many finer powders within WCP, which all have a larger specific surface area, and we have included reference objects in the text.

Revision: Compared with silicate cement, WCP itself has higher water absorption and water demand, which will adversely affect the mechanical and working properties of cementitious materials, and this effect will become larger with the increase of WCP admixture.

Point 4: Page 3, under ‘3. Activation of WCP’, How does WCP reduce the energy of cement production when it is further activated by extra chemical/thermal or mechanical methods?

Reply: The production of cement requires a high temperature of 1450℃ burning, which produces a lot of energy and produces a lot of carbon dioxide. The three activation energy consumption of WCP is not as high as that of cement production, and the energy consumption of mechanical activation is less mainly through ball mill grinding. Chemical activation mainly uses chemical reagents to provide an alkaline environment with minimal energy consumption. Thermal activation energy consumption is high, but the highest temperature is only about 800℃, far less than the 1450℃ high temperature required for cement production.

Revision: We have added the feasibility analysis of the three methods in Table 5 of 3.4.

Point 5: Under the same topic, you suddenly introduced a mortar prepared by replacing 30% cement, which mortar are you refereeing to?

Reply: We refer to the method of determining WCP activity index in JG/T573-2020 "Recycled Micronized Powder for Concrete and Mortar", using WCP instead of 30% cement for the preparation of colloidal sand, after curing for 28 days, the compressive strength after 28 days is measured and compared with the mortar without WCP, that is, the activity index (SAI) of WCP is obtained, SAI>60 is Grade I recycled micronized powder, 60 < SAI<70 is grade II recycled micronized powder. The colloidal sand in which was prepared by the standard sand used.

Revision: For the reader's understanding, we have marked the reference sources where relevant.

Point 6: Under 3.1 Mechanical activation, you have mentioned the term ‘volcanic ash activity’. I suppose volcanic ash is different from WCP and how do you relate it with WCP?

Reply: Thank you for pointing out the error for us, 'volcanic ash activity' is indeed inappropriate to use here and we have made a change to the sentence.

Revision: However, WCP should not be ground too finely, otherwise it will lead to agglomeration of the material and reduce the strength of the specimen.

Point 7: On page 6, referring to the sentence “When the maintenance period was extended to 28 days, the strength of the regenerated mortar gradually decreased with increasing Ca(OH)2 admixture and was less than 0” … do you mean the strength goes gradually to 0 or the difference in strength goes to 0? The next statement also doesn’t sound good. Because the compressive strength at an early age increased and later at 28 days it decreased, but the reason given is for an increase in compressive strength at later ages. Please look at your wording and try to rephrase it.

Reply: Thank you for helping us point out the error, which we have corrected for better understanding by our readers.

Revision: Related scholars [32,41] found that some Ca(OH)2 could not significantly improve the compressive and flexural strength of the specimens at the early stage (3 and 7 days). However, when the age of maintenance was extended to 28 d, the strength of the recycled mortar was significantly improved. Some scholars [42-43] found that when Ca(OH)2 was dosed at 1%, the compressive strength at the age of 3d and 7d decreased, and the strength at the age of 28d increased, which was because Ca(OH)2 could not dissolve rapidly to occur hydration reaction in the early stage, but could react with 3CaO-Al2O3 in cement in the later stage to form a large amount of C3AH6, and then increase the strength of mortar specimens. Mixing 1% Ca(OH)2 exciter can effectively excite the activity of WCP, and its activity index reaches the highest value of 64.32, see Fig. 9. The reactions of active SiO2, Al2O3 and Ca(OH)2 with water are shown in (2) (3).

Point 8: Figure 6, what is the difference in activation using CaO and Ca(OH)2? in an aqueous solution CaO exists in the form of Ca(OH)2. Is CaO used in solid form?

Reply: CaO is added as a solid, and the difference between it and Ca(OH)2 is that CaO gives off a lot of heat when it meets water, while Ca(OH)2 is slightly soluble in water and does not give off such a large amount of heat, so the pre-product strength of CaO-activated WCP will

Revision: We have explained this in the relevant sections for better understanding.

Point 9: Full stops missing here and there through the text. Try to carefully read the article to address that.

Reply: We double-checked the manuscript and the article does have the problem of too many long sentences, and we have tried to make some changes. By the way, this manuscript submitted to MDPI for English editing has been edited by a professional company. The final result can be seen in the revised article.

Point 10: Under 4.2 Mechanical properties…’coagulation time’ is not a very common term in concrete research. Do you mean the setting time of mortar/concrete?

Reply: Thank you for your suggestion. setting time is indeed better than coagulation time, and we have already made changes in the original text.

Revision: 4.2.1. setting time

Point 11: The title 4.2.2 Fluidity, can be replaced with common terms in concrete/mortar research such as consistency or workability.

Reply: Thank you for your valuable advice, workability is indeed more professional than Fluidity, and we have amended this in the original article.

Revision: 4.2.2. Workability

Round 2

Reviewer 1 Report

Comments

Figure 3 and Figure 4 - Image quality has to be improved for better Viewability and readability

Author Response

Response to the comments on Manuscript ID sustainability-2234874

Dear Editor and the Reviewers:

We would like to thank for thoroughly reviewing our manuscript and making many thoughtful comments. These comments helped us improve our study and provided paramount guidance for future research.

We have addressed the comments to the best of our abilities and some necessary explanations are added. In order to easily find the modified parts, these parts are marked with color in the revised manuscript. And our responses point-by-point are listed as follows.

Hope to hear from you soon.

Yours sincerely,

Baolin Tan

Reviewer 3 Report

The author has seriously responded to all the comments made by the reviewers, and the quality of the article has been improved. It is recommended to accept the paper. In addition, the author is requested to check the manuscript carefully. There are some minor errors, such as Chinese, in the reply comments.

Author Response

Response to the comments on Manuscript ID sustainability-2234874

Dear Editor and the Reviewers:

We would like to thank you for thoroughly reviewing our manuscript and making many thoughtful comments. These comments helped us improve our study and provided paramount guidance for future research.

We have addressed the comments to the best of our abilities and some necessary explanations are added. To easily find the modified parts, these parts are marked with color in the revised manuscript. And our responses point-by-point are listed as follows.

Hope to hear from you soon.

Yours sincerely,

Baolin Tan

Response to Reviewer 3 Comments

Comment: The author has seriously responded to all the comments made by the reviewers, and the quality of the article has been improved. It is recommended to accept the paper. In addition, the author is requested to check the manuscript carefully. There are some minor errors, such as in Chinese, in the reply comments.

Reply: Sorry, due to our carelessness, you had a bad reading experience, we have checked and corrected the wrong part of the reply and modified the part with Chinese.

Revision: Here are the comments after our revision.
